# Multitask Spectral Learning of Weighted Automata

**Guillaume Rabusseau** *
McGill University

**Borja Balle** †
Amazon Research Cambridge

**Joelle Pineau**‡
McGill University

## Abstract

We consider the problem of estimating multiple related functions computed by weighted automata (WFA). We first present a natural notion of relatedness between WFAs by considering to which extent several WFAs can share a common underlying representation. We then introduce the novel model of vector-valued WFA which conveniently helps us formalize this notion of relatedness. Finally, we propose a spectral learning algorithm for vector-valued WFAs to tackle the multitask learning problem. By jointly learning multiple tasks in the form of a vector-valued WFA, our algorithm enforces the discovery of a representation space shared between tasks. The benefits of the proposed multitask approach are theoretically motivated and showcased through experiments on both synthetic and real world datasets.

## 1 Introduction

One common task in machine learning consists in estimating an unknown function $f : \mathcal{X} \to \mathcal{Y}$ from a training sample of input-output data $\{(x_i, y_i)\}_{i=1}^{N}$ where each $y_i \simeq f(x_i)$ is a (possibly noisy) estimate of $f(x_i)$. In *multitask learning*, the learner is given several such learning tasks $f_1, \cdots, f_m$. It has been shown, both experimentally and theoretically, that learning related tasks simultaneously can lead to better performances relative to learning each task independently (see e.g. [1, 7], and references therein). Multitask learning has proven particularly useful when few data points are available for each task, or when it is difficult or costly to collect data for a target task while much data is available for related tasks (see e.g. [28] for an example in healthcare). In this paper, we propose a multitask learning algorithm for the case where the input space $\mathcal{X}$ consists of *sequence data*.

Many tasks in natural language processing, computational biology, or reinforcement learning, rely on estimating functions mapping sequences of observations to real numbers: e.g. inferring probability distributions over sentences in language modeling or learning the dynamics of a model of the environment in reinforcement learning. In this case, the function $f$ to infer from training data is defined over the set $\Sigma^*$ of strings built on a finite alphabet $\Sigma$. *Weighted finite automata* (WFA) are finite state machines that allow one to succinctly represent such functions. In particular, WFAs can compute any probability distribution defined by a hidden Markov model (HMM) [13] and can model the transition and observation behavior of partially observable Markov decision processes [26]. A recent line of work has led to the development of *spectral methods* for learning HMMs [17], WFAs [2, 4] and related models, offering an alternative to EM based algorithms with the benefits of being computationally efficient and providing consistent estimators. Spectral learning algorithms have led to competitive results in the fields of natural language processing [12, 3] and robotics [8].

We consider the problem of multitask learning for WFAs. As a motivational example, consider a natural language modeling task where one needs to make predictions in different contexts (e.g. online chat vs. newspaper articles) and has access to datasets in each of them; it is natural to expect that basic grammar is shared across the datasets and that one could benefit from simultaneously

learning these tasks. The notion of relatedness between tasks can be expressed in different ways; one common assumption in multitask learning is that the multiple tasks share a *common underlying representation* [6, 11]. In this paper, we present a natural notion of shared representation between functions defined over strings and we propose a learning algorithm that encourages the discovery of this shared representation. Intuitively, our notion of relatedness captures to which extent several functions can be computed by WFAs sharing a joint forward feature map. In order to formalize this notion of relatedness, we introduce the novel model of *vector-valued WFA* (vv-WFA) which generalizes WFAs to vector-valued functions and offer a natural framework to formalize the multitask learning problem. Given $m$ tasks $f_1, \cdots, f_m : \Sigma^* \to \mathbb{R}$, we consider the function $\vec{f} = [f_1, \cdots, f_m] : \Sigma^* \to \mathbb{R}^m$ whose output for a given input string $x$ is the $m$-dimensional vector having entries $f_i(x)$ for $i = 1, \cdots, m$. We show that the notion of *minimal vv-WFA* computing $\vec{f}$ exactly captures our notion of relatedness between tasks and we prove that the dimension of such a minimal representation is equal to the rank of a flattening of the *Hankel tensor* of $\vec{f}$ (Theorem 3). Leveraging this result, we design a spectral learning algorithm for vv-WFAs which constitutes a sound multitask learning algorithm for WFAs: by learning $\vec{f}$ in the form of a vv-WFA, rather than independently learning a WFA for each task $f_i$, we implicitly enforce the discovery of a joint feature space shared among all tasks. After giving a theoretical insight on the benefits of this multitask approach (by leveraging a recent result on asymmetric bounds for singular subspace estimation [9]), we conclude by showcasing these benefits with experiments on both synthetic and real world data.

**Related work**. Multitask learning for sequence data has previously received limited attention. In [16], mixtures of Markov chains are used to model dynamic user profiles. Tackling the multitask problem with nonparametric Bayesian methods is investigated in [15] to model related time series with Beta processes and in [23] to discover relationships between related datasets using nested Dirichlet process and infinite HMMs. Extending recurrent neural networks to the multitask setting has also recently received some interest (see e.g. [21, 22]). To the best of our knowledge, this paper constitutes the first attempt to tackle the multitask problem for the class of functions computed by general WFAs.

## 2   Preliminaries

We first present notions on weighted automata, spectral learning of weighted automata and tensors. We start by introducing some notation. We denote by $\Sigma^*$ the set of strings on a finite alphabet $\Sigma$. The empty string is denoted by $\lambda$ and the length of a string $x$ by $|x|$. For any integer $k$ we let $[k] = \{1, 2, \cdots, k\}$. We use lower case bold letters for vectors (e.g. $\mathbf{v} \in \mathbb{R}^{d_1}$), upper case bold letters for matrices (e.g. $\mathbf{M} \in \mathbb{R}^{d_1 \times d_2}$) and bold calligraphic letters for higher order tensors (e.g. $\boldsymbol{\mathcal{T}} \in \mathbb{R}^{d_1 \times d_2 \times d_3}$). The $i$th row (resp. column) of a matrix $\mathbf{M}$ will be denoted by $\mathbf{M}_{i,:}$ (resp. $\mathbf{M}_{:,i}$). This notation is extended to slices of a tensor in the straightforward way. Given a matrix $\mathbf{M} \in \mathbb{R}^{d_1 \times d_2}$, we denote by $\mathbf{M}^\dagger$ its Moore-Penrose pseudo-inverse and by $\text{vec}(\mathbf{M}) \in \mathbb{R}^{d_1 d_2}$ its vectorization.

**Weighted finite automaton**. A *weighted finite automaton* (WFA) with $n$ states is a tuple $A = (\boldsymbol{\alpha}, \{\mathbf{A}^\sigma\}_{\sigma \in \Sigma}, \boldsymbol{\omega})$ where $\boldsymbol{\alpha}, \boldsymbol{\omega} \in \mathbb{R}^n$ are the initial and final weights vectors respectively, and $\mathbf{A}^\sigma \in \mathbb{R}^{n \times n}$ is the transition matrix for each symbol $\sigma \in \Sigma$. A WFA computes a function $f_A : \Sigma^* \to \mathbb{R}$ defined for each word $x = x_1 x_2 \cdots x_k \in \Sigma^*$ by $f_A(x) = \boldsymbol{\alpha}^\top \mathbf{A}^{x_1} \mathbf{A}^{x_2} \cdots \mathbf{A}^{x_k} \boldsymbol{\omega}$.

By letting $\mathbf{A}^x = \mathbf{A}^{x_1} \mathbf{A}^{x_2} \cdots \mathbf{A}^{x_k}$ for any word $x = x_1 x_2 \cdots x_k \in \Sigma^*$ we will often use the shorter notation $f_A(x) = \boldsymbol{\alpha}^\top \mathbf{A}^x \boldsymbol{\omega}$. A WFA $A$ with $n$ states is *minimal* if its number of states is minimal, i.e. any WFA $B$ such that $f_A = f_B$ has at least $n$ states. A function $f : \Sigma^* \to \mathbb{R}$ is *recognizable* if it can be computed by a WFA. In this case the *rank* of $f$ is the number of states of a minimal WFA computing $f$, if $f$ is not recognizable we let $\text{rank}(f) = \infty$.

**Hankel matrix**. The *Hankel matrix* $\mathbf{H}_f \in \mathbb{R}^{\Sigma^* \times \Sigma^*}$ associated with a function $f : \Sigma^* \to \mathbb{R}$ is the infinite matrix with entries $(\mathbf{H}_f)_{u,v} = f(uv)$ for $u, v \in \Sigma^*$. The spectral learning algorithm for WFAs relies on the following fundamental relation between the rank of $f$ and the rank of $\mathbf{H}_f$.

**Theorem 1.** *[10, 14] For any function $f : \Sigma^* \to \mathbb{R}$, $\text{rank}(f) = \text{rank}(\mathbf{H}_f)$.*

**Spectral learning**. Showing that the rank of the Hankel matrix is upper bounded by the rank of $f$ is easy: given a WFA $A = (\boldsymbol{\alpha}, \{\mathbf{A}^\sigma\}_{\sigma \in \Sigma}, \boldsymbol{\omega})$ with $n$ states, we have the rank $n$ factorization $\mathbf{H}_f = \mathbf{PS}$ where the matrices $\mathbf{P} \in \mathbb{R}^{\Sigma^* \times n}$ and $\mathbf{S} \in \mathbb{R}^{n \times \Sigma^*}$ are defined by $\mathbf{P}_{u,:} = \boldsymbol{\alpha}^\top \mathbf{A}^u$ and $\mathbf{S}_{:,v} = \mathbf{A}^v \boldsymbol{\omega}$ for

all $u, v \in \Sigma^*$. The converse is more tedious to show but its proof is constructive, in the sense that it allows one to build a WFA computing $f$ from any rank $n$ factorization of $\mathbf{H}_f$. This construction is the cornerstone of the spectral learning algorithm and is given in the following corollary.

**Corollary 2.** *[4, Lemma 4.1] Let $f : \Sigma^* \to \mathbb{R}$ be a recognizable function with rank $n$, let $\mathbf{H} \in \mathbb{R}^{\Sigma^* \times \Sigma^*}$ be its Hankel matrix, and for each $\sigma \in \Sigma$ let $\mathbf{H}^\sigma \in \mathbb{R}^{\Sigma^* \times \Sigma^*}$ be defined by $\mathbf{H}^\sigma_{u,v} = f(u\sigma v)$ for all $u, v \in \Sigma^*$.*

*Then, for any $\mathbf{P} \in \mathbb{R}^{\Sigma^* \times n}$, $\mathbf{S} \in \mathbb{R}^{n \times \Sigma^*}$ such that $\mathbf{H} = \mathbf{PS}$, the WFA $A = (\boldsymbol{\alpha}, \{\mathbf{A}^\sigma\}_{\sigma \in \Sigma}, \boldsymbol{\omega})$ where $\boldsymbol{\alpha}^\top = \mathbf{P}_{\lambda,:}$, $\boldsymbol{\omega} = \mathbf{S}_{:,\lambda}$, and $\mathbf{A}^\sigma = \mathbf{P}^\dagger \mathbf{H}^\sigma \mathbf{S}^\dagger$ is a minimal WFA for $f$.*

In practice, finite sub-blocks of the Hankel matrices are used. Given finite sets of prefixes and suffixes $\mathcal{P}, \mathcal{S} \subset \Sigma^*$, let $\mathbf{H}_{\mathcal{P},\mathcal{S}}, \{\mathbf{H}^\sigma_{\mathcal{P},\mathcal{S}}\}_{\sigma \in \Sigma}$ be the finite sub-blocks of $\mathbf{H}$ whose rows (resp. columns) are indexed by prefixes in $\mathcal{P}$ (resp. suffixes in $\mathcal{S}$). One can show that if $\mathcal{P}$ and $\mathcal{S}$ are such that $\lambda \in \mathcal{P} \cap \mathcal{S}$ and $\mathrm{rank}(\mathbf{H}) = \mathrm{rank}(\mathbf{H}_{\mathcal{P},\mathcal{S}})$, then the previous corollary still holds, i.e. a minimal WFA computing $f$ can be recovered from any rank $n$ factorization of $\mathbf{H}_{\mathcal{P},\mathcal{S}}$. The spectral method thus consists in estimating the matrices $\mathbf{H}_{\mathcal{P},\mathcal{S}}, \mathbf{H}^\sigma_{\mathcal{P},\mathcal{S}}$ from training data (using e.g. empirical frequencies if $f$ is stochastic), finding a low-rank factorization of $\mathbf{H}_{\mathcal{P},\mathcal{S}}$ (using e.g. SVD) and constructing a WFA approximating $f$ using Corollary 2.

**Tensors**. We make a sporadic use of tensors in this paper, we thus introduce the few necessary definitions and notations; more details can be found in [18]. A *3rd order tensor* $\boldsymbol{\mathcal{T}} \in \mathbb{R}^{d_1 \times d_2 \times d_3}$ can be seen as a multidimensional array $(\boldsymbol{\mathcal{T}}_{i_1,i_2,i_3} : i_1 \in [d_1], i_2 \in [d_2], , i_3 \in [d_3])$. The *mode-$n$ fibers* of $\boldsymbol{\mathcal{T}}$ are the vectors obtained by fixing all indices except the $n$th one, e.g. $\boldsymbol{\mathcal{T}}_{:,i_2,i_3} \in \mathbb{R}^{d_1}$. The *$n$th mode flattening* of $\boldsymbol{\mathcal{T}}$ is the matrix having the mode-$n$ fibers of $\boldsymbol{\mathcal{T}}$ for columns and is denoted by e.g. $\boldsymbol{\mathcal{T}}_{(1)} \in \mathbb{R}^{d_1 \times d_2 d_3}$. The *mode-1 matrix product* of a tensor $\boldsymbol{\mathcal{T}} \in \mathbb{R}^{d_1 \times d_2 \times d_3}$ and a matrix $\mathbf{X} \in \mathbb{R}^{m \times d_1}$ is a tensor of size $m \times d_2 \times d_3$ denoted by $\boldsymbol{\mathcal{T}} \times_1 \mathbf{X}$ and defined by the relation $\boldsymbol{\mathcal{Y}} = \boldsymbol{\mathcal{T}} \times_1 \mathbf{X} \Leftrightarrow \boldsymbol{\mathcal{Y}}_{(1)} = \mathbf{X} \boldsymbol{\mathcal{T}}_{(1)}$; the mode-$n$ product for $n = 2, 3$ is defined similarly.

# 3   Vector-Valued WFAs for Multitask Learning

In this section, we present a notion of *relatedness between WFAs* that we formalize by introducing the novel model of *vector-valued weighted automaton*. We then propose a multitask learning algorithm for WFAs by designing a spectral learning algorithm for vector-valued WFAs.

**A notion of relatedness between WFAs**. The basic idea behind our approach emerges from interpreting the computation of a WFA as a linear model in some feature space. Indeed, the computation of a WFA $A = (\boldsymbol{\alpha}, \{\mathbf{A}^\sigma\}_{\sigma \in \Sigma}, \boldsymbol{\omega})$ with $n$ states on a word $x \in \Sigma^*$ can be seen as first mapping $x$ to an $n$-dimensional feature vector through a *compositional feature map* $\phi : \Sigma^* \to \mathbb{R}^n$, and then applying a linear form in the feature space to obtain the final value $f_A(x) = \langle \phi(x), \boldsymbol{\omega} \rangle$. The feature map is defined by $\phi(x)^\top = \boldsymbol{\alpha}^\top \mathbf{A}^x$ for all $x \in \Sigma^*$ and it is compositional in the sense that for any $x \in \Sigma^*$ and any $\sigma \in \Sigma$ we have $\phi(x\sigma)^\top = \phi(x)^\top \mathbf{A}^\sigma$. We will say that such a feature map is *minimal* if the linear space $V \subset \mathbb{R}^n$ spanned by the vectors $\{\phi(x)\}_{x \in \Sigma^*}$ is of dimension $n$. Theorem 1 implies that the dimension of $V$ is actually equal to the rank of $f_A$, showing that the notion of minimal feature map naturally coincides with the notion of minimal WFA.

A notion of *relatedness between WFAs* naturally arises by considering to which extent two (or more) WFAs can share a joint feature map $\phi$. More precisely, consider two recognizable functions $f_1, f_2 : \Sigma^* \to \mathbb{R}$ of rank $n_1$ and $n_2$ respectively, with corresponding feature maps $\phi_1 : \Sigma^* \to \mathbb{R}^{n_1}$ and $\phi_2 : \Sigma^* \to \mathbb{R}^{n_2}$. Then, a joint feature map for $f_1$ and $f_2$ always exists and is obtained by considering the direct sum $\phi_1 \oplus \phi_2 : \Sigma^* \to \mathbb{R}^{n_1 + n_2}$ that simply concatenates the feature vectors $\phi_1(x)$ and $\phi_2(x)$ for any $x \in \Sigma^*$. However, this feature map may not be minimal, i.e. there may exist another joint feature map of dimension $n < n_1 + n_2$. Intuitively, the smaller this minimal dimension $n$ is the more related the two tasks are, with the two extremes being on the one hand $n = n_1 + n_2$ where the two tasks are independent, and on the other hand e.g. $n = n_1$ where one of the (minimal) feature maps $\phi_1, \phi_2$ is sufficient to predict both tasks.

**Vector-valued WFA**. We now introduce a computational model for vector-valued functions on strings that will help formalize this notion of relatedness between WFAs.

**Definition 1.** *A $d$-dimensional* vector-valued weighted finite automaton *(vv-WFA) with $n$ states is a tuple $A = (\boldsymbol{\alpha}, \{\mathbf{A}^\sigma\}_{\sigma \in \Sigma}, \boldsymbol{\Omega})$ where $\boldsymbol{\alpha} \in \mathbb{R}^n$ is the initial weights vector, $\boldsymbol{\Omega} \in \mathbb{R}^{n \times d}$ is the matrix of final weights, and $\mathbf{A}^\sigma \in \mathbb{R}^{n \times n}$ is the transition matrix for each symbol $\sigma \in \Sigma$. A vv-WFA computes a function $\vec{f}_A : \Sigma^* \to \mathbb{R}^d$ defined by*

$$\vec{f}_A(x) = \boldsymbol{\alpha}^\top \mathbf{A}^{x_1} \mathbf{A}^{x_2} \cdots \mathbf{A}^{x_k} \boldsymbol{\Omega}$$

*for each word $x = x_1 x_2 \cdots x_k \in \Sigma^*$.*

We extend the notions of recognizability, minimality and rank of a WFA in the straightforward way: a function $\vec{f} : \Sigma^* \to \mathbb{R}^d$ is recognizable if it can be computed by a vv-WFA, a vv-WFA is minimal if its number of states is minimal, and the rank of $\vec{f}$ is the number of states of a minimal vv-WFA computing $\vec{f}$. A $d$-dimensional vv-WFA can be seen as a collection of $d$ WFAs that all share their initial vectors and transition matrices but have different final vectors. Alternatively, one could take a dual approach and define vv-WFAs as a collection of WFAs sharing transitions and final vectors[4].

**vv-WFAs and relatedness between WFAs**. We now show how the vv-WFA model naturally captures the notion of relatedness presented above. Recall that this notion intends to capture to which extent two recognizable functions $f_1, f_2 : \Sigma^* \to \mathbb{R}$, of ranks $n_1$ and $n_2$ respectively, can share a joint forward feature map $\phi : \Sigma^* \to \mathbb{R}^n$ satisfying $f_1(x) = \langle \phi(x), \boldsymbol{\omega}_1 \rangle$ and $f_2(x) = \langle \phi(x), \boldsymbol{\omega}_2 \rangle$ for all $x \in \Sigma^*$, for some $\boldsymbol{\omega}_1, \boldsymbol{\omega}_2 \in \mathbb{R}^n$. Consider the vector-valued function $\vec{f} = [f_1, f_2] : \Sigma^* \to \mathbb{R}^2$ defined by $\vec{f}(x) = [f_1(x), f_2(x)]$ for all $x \in \Sigma^*$. It can easily be seen that the minimal dimension of a shared forward feature map between $f_1$ and $f_2$ is exactly the rank of $\vec{f}$, i.e. the number of states of a minimal vv-WFA computing $\vec{f}$. This notion of relatedness can be generalized to more than two functions by considering $\vec{f} = [f_1, \cdots, f_m]$ for $m$ different recognizable functions $f_1, \cdots, f_m$ of respective ranks $n_1, \cdots, n_m$. In this setting, it is easy to check that the rank of $\vec{f}$ lies between $\max(n_1, \cdots, n_m)$ and $n_1 + \cdots + n_m$; smaller values of this rank leads to a smaller dimension of the minimal forward feature map and thus, intuitively, to more closely related tasks. We now formalize this measure of relatedness between recognizable functions.

**Definition 2.** *Given $m$ recognizable functions $f_1, \cdots, f_m$, we define their* relatedness measure *by $\tau(f_1, \cdots, f_m) = 1 - (\mathrm{rank}(\vec{f}) - \max_i \mathrm{rank}(f_i))/\sum_i \mathrm{rank}(f_i)$ where $\vec{f} = [f_1, \cdots, f_m]$.*

One can check that this measure of relatedness takes its values in $(0, 1]$. We say that tasks are *maximally related* when their relatedness measure is 1 and *independent* when it is minimal. Observe that the rank $R$ of a vv-WFA does not give enough information to determine whether one set of tasks is more related than another: the degree of relatedness depends on the relation between $R$ and the ranks of each individual task. The relatedness parameter $\tau$ circumvents this issue by measuring where $R$ stands between the maximum rank over the different tasks and the sum of their ranks.

**Example 1.** *Let $\Sigma = \{a, b, c\}$ and let $|x|_\sigma$ denotes the number of occurrences of $\sigma$ in $x$ for any $\sigma \in \Sigma$. Consider the functions defined by $f_1(x) = 0.5|x|_a + 0.5|x|_b$, $f_2(x) = 0.3|x|_b - 0.6|x|_c$ and $f_3(x) = |x|_c$ for all $x \in \Sigma^*$. It is easy to check that $\mathrm{rank}(f_1) = \mathrm{rank}(f_2) = 4$ and $\mathrm{rank}(f_3) = 2$. Moreover, $f_2$ and $f_3$ are maximally related (indeed $\mathrm{rank}([f_2, f_3]) = 4 = \mathrm{rank}(f_2)$ thus $\tau(f_2, f_3) = 1$), $f_1$ and $f_3$ are independent (indeed $\tau(f_1, f_3) = 2/3$ is minimal since $\mathrm{rank}([f_1, f_3]) = 6 = \mathrm{rank}(f_1) + \mathrm{rank}(f_3)$), and $f_1$ and $f_2$ are related but not maximally related (since $4 = \mathrm{rank}(f_1) = \mathrm{rank}(f_2) < \mathrm{rank}([f_1, f_2]) = 6 < \mathrm{rank}(f_1) + \mathrm{rank}(f_2) = 8$).*

**Spectral learning of vv-WFAs**. We now design a spectral learning algorithm for vv-WFAs. Given a function $\vec{f} : \Sigma^* \to \mathbb{R}^d$, we define its Hankel tensor $\boldsymbol{\mathcal{H}} \in \mathbb{R}^{\Sigma^* \times d \times \Sigma^*}$ by $\boldsymbol{\mathcal{H}}_{u,:,v} = \vec{f}(uv)$ for all $u, v \in \Sigma^*$. We first show in Theorem 3 (whose proof can be found in the supplementary material) that the fundamental relation between the rank of a function and the rank of its Hankel matrix can naturally be extended to the vector-valued case. Compared with Theorem 1, the Hankel matrix is now replaced by the mode-1 flattening $\boldsymbol{\mathcal{H}}_{(1)}$ of the Hankel tensor (which can be obtained by concatenating the matrices $\boldsymbol{\mathcal{H}}_{:,i,:}$ along the horizontal axis).

**Theorem 3** (Vector-valued Fliess Theorem). *Let $\vec{f} : \Sigma^* \to \mathbb{R}^d$ and let $\boldsymbol{\mathcal{H}}$ be its Hankel tensor. Then $\mathrm{rank}(\vec{f}) = \mathrm{rank}(\boldsymbol{\mathcal{H}}_{(1)})$.*

Similarly to the scalar-valued case, this theorem can be leveraged to design a spectral learning algorithm for vv-WFAs. The following corollary (whose proof can be found in the supplementary material) shows how a vv-WFA computing a recognizable function $\vec{f} : \Sigma^* \to \mathbb{R}^d$ of rank $n$ can be recovered from any rank $n$ factorization of its Hankel tensor.

**Corollary 4.** *Let $\vec{f} : \Sigma^* \to \mathbb{R}^d$ be a recognizable function with rank $n$, let $\mathcal{H} \in \mathbb{R}^{\Sigma^* \times d \times \Sigma^*}$ be its Hankel tensor, and for each $\sigma \in \Sigma$ let $\mathcal{H}^\sigma \in \mathbb{R}^{\Sigma^* \times d \times \Sigma^*}$ be defined by $\mathcal{H}^\sigma_{u,:,v} = \vec{f}(u\sigma v)$ for all $u, v \in \Sigma^*$.*

*Then, for any $\mathbf{P} \in \mathbb{R}^{\Sigma^* \times n}$ and $\mathcal{S} \in \mathbb{R}^{n \times d \times \Sigma^*}$ such that $\mathcal{H} = \mathcal{S} \times_1 \mathbf{P}$, the vv-WFA $A = (\boldsymbol{\alpha}, \{\mathbf{A}^\sigma\}_{\sigma \in \Sigma}, \boldsymbol{\Omega})$ defined by $\boldsymbol{\alpha}^\top = \mathbf{P}_{\lambda,:}$, $\boldsymbol{\Omega} = \mathcal{S}_{:,:,\lambda}$, and $\mathbf{A}^\sigma = \mathbf{P}^\dagger \mathcal{H}^\sigma_{(1)}(\mathcal{S}_{(1)})^\dagger$ is a minimal vv-WFA computing $\vec{f}$.*

Similarly to the scalar-valued case, one can check that the previous corollary also holds for any finite sub-tensors $\mathcal{H}_{\mathcal{P},\mathcal{S}}, \{\mathcal{H}^\sigma_{\mathcal{P},\mathcal{S}}\}_{\sigma \in \Sigma}$ of $\mathcal{H}$ indexed by prefixes and suffixes in $\mathcal{P}, \mathcal{S} \subset \Sigma^*$, whenever $\mathcal{P}$ and $\mathcal{S}$ are such that $\lambda \in \mathcal{P} \cap \mathcal{S}$ and $\text{rank}(\mathcal{H}_{(1)}) = \text{rank}((\mathcal{H}_{\mathcal{P},\mathcal{S}})_{(1)})$; we will call such a basis $(\mathcal{P}, \mathcal{S})$ *complete*. The spectral learning algorithm for vv-WFAs then consists in estimating these Hankel tensors from training data and using Corollary 4 to recover a vv-WFA approximating the target function. Of course a noisy estimate of the Hankel tensor $\hat{\mathcal{H}}$ will not be of low rank and the factorization $\hat{\mathcal{H}} = \mathcal{S} \times_1 \mathbf{P}$ should only be performed approximately in order to counter the presence of noise. In practice a low rank approximation of $\hat{\mathcal{H}}_{(1)}$ is obtained using truncated SVD.

**Multitask learning of WFAs.** Let us now go back to the multitask learning problem and let $f_1, \cdots f_m : \Sigma^* \to \mathbb{R}$ be multiple functions we wish to infer in the form of WFAs. The spectral learning algorithm for vv-WFAs naturally suggests a way to tackle this multitask problem: by learning $\vec{f} = [f_1, \cdots, f_m]$ in the form of a vv-WFA, rather than independently learning a WFA for each task $f_i$, we implicitly enforce the discovery of a joint forward feature map shared among all tasks.

We will now see how a further step can be added to this learning scheme to enforce more robustness to noise. The motivation for this additional step comes from the observation that even though a $d$-dimensional vv-WFA $A = (\boldsymbol{\alpha}, \{\mathbf{A}^\sigma\}_{\sigma \in \Sigma}, \boldsymbol{\Omega})$ may be minimal, the corresponding scalar-valued WFAs $A_i = \langle \boldsymbol{\alpha}, \{\mathbf{A}^\sigma\}_{\sigma \in \Sigma}, \boldsymbol{\Omega}_{:,i} \rangle$ for $i \in [d]$ may not be. Suppose for example that $A_1$ is not minimal. This implies that some part of its state space does not contribute to the function $f_1$ but comes from asking for a rich enough state representation that can predict other tasks as well. Moreover, when one learns a vv-WFA from noisy estimates of the Hankel tensors, the rank $R$ approximation $\hat{\mathcal{H}}_{(1)} \simeq \mathbf{P} \mathcal{S}_{(1)}$ somehow annihilates the noise contained in the space orthogonal to the top $R$ singular vectors of $\hat{\mathcal{H}}_{(1)}$, but when the WFA $A_1$ has rank $R_1 < R$ we intuitively see that there is still a subspace of dimension $R - R_1$ containing only irrelevant features. In order to circumvent this issue, we would like to project down the (scalar-valued) WFAs $A_i$ down to their true dimensions, intuitively enforcing each predictor to use as few features as possible for each task, and thus annihilating the noise lying in the corresponding irrelevant subspaces. To achieve this we will make use of the following proposition that explicits the projections needed to obtain minimal scalar-valued WFAs from a given vv-WFA (the proof is given in the supplementary material).

**Proposition 1.** *Let $\vec{f} : \Sigma^* \to \mathbb{R}^d$ be a function computed by a minimal vv-WFA $A = (\boldsymbol{\alpha}, \{\mathbf{A}^\sigma\}_{\sigma \in \Sigma}, \boldsymbol{\Omega})$ with $n$ states and let $\mathcal{P}, \mathcal{S} \subseteq \Sigma^*$ be a complete basis for $\vec{f}$. For any $i \in [d]$, let $f_i : \Sigma^* \to \mathbb{R}$ be defined by $f_i(x) = \vec{f}(x)_i$ for all $x \in \Sigma^*$ and let $n_i$ denote the rank of $f_i$.*

*Let $\mathbf{P} \in \mathbb{R}^{\mathcal{P} \times n}$ be defined by $\mathbf{P}_{x,:} = \boldsymbol{\alpha}^\top \mathbf{A}^x$ for all $x \in \mathcal{P}$ and, for $i \in [d]$, let $\mathbf{H}_i \in \mathbb{R}^{\mathcal{P} \times \mathcal{S}}$ be the Hankel matrix of $f_i$ and let $\mathbf{H}_i = \mathbf{U}_i \mathbf{D}_i \mathbf{V}_i^\top$ be its thin SVD (i.e. $\mathbf{D}_i \in \mathbb{R}^{n_i \times n_i}$).*

*Then, for any $i \in [d]$, the WFA $A_i = \langle \boldsymbol{\alpha}_i, \{\mathbf{A}^\sigma_i\}_{\sigma \in \Sigma}, \boldsymbol{\omega}_i \rangle$ defined by*

$$\boldsymbol{\alpha}_i^\top = \boldsymbol{\alpha}^\top \mathbf{P}^\dagger \mathbf{U}_i, \quad \boldsymbol{\omega}_i = \mathbf{U}_i^\top \mathbf{P} \boldsymbol{\Omega}_{:,i} \text{ and } \mathbf{A}^\sigma_i = \mathbf{U}_i^\top \mathbf{P} \mathbf{A}^\sigma \mathbf{P}^\dagger \mathbf{U}_i \text{ for each } \sigma \in \Sigma,$$

*is a minimal WFA computing $f_i$.*

Given noisy estimates $\hat{\mathcal{H}}, \{\hat{\mathcal{H}}^\sigma\}_{\sigma \in \Sigma}$ of the Hankel tensors of a function $\vec{f}$ and estimates $R$ of the rank of $\vec{f}$ and $R_i$ of the ranks of the $f_i$'s, the first step of the learning algorithm consists in applying Corollary 4 to the factorization $\hat{\mathcal{H}}_{(1)} \simeq \mathbf{U}(\mathbf{DV}^\top)$ obtained by truncated SVD to get a

vv-WFA $A$ approximating $\vec{f}$. Then, Proposition 1 can be used to project down each WFA $A_i$ by estimating $\mathbf{U}_i$ with the top $R_i$ left singular vectors of $\hat{\boldsymbol{\mathcal{H}}}_{:,i,:}$. The overall procedure for our Multi-Task Spectral Learning (MT-SL) is summarized in Algorithm 1 where lines 1-3 correspond to the vv-WFA estimation while lines 4-7 correspond to projecting down the corresponding scalar-valued WFAs. To further motivate the projection step, let us consider the case when $m$ tasks are completely unrelated, and each of them requires $n$ states. Single-task learning would lead to a model with $\mathcal{O}\left(|\Sigma|mn^2\right)$ parameters, while the multi-task learning approach would return a larger model of size $\mathcal{O}\left(|\Sigma|(mn)^2\right)$; the projection step eliminates such redundancy.

---

**Algorithm 1** `MT-SL`: Spectral Learning of vector-valued WFA for multitask learning

---

**Input:** Empirical Hankel tensors $\hat{\boldsymbol{\mathcal{H}}}, \{\hat{\boldsymbol{\mathcal{H}}}^\sigma\}_{\sigma \in \Sigma}$ of size $\mathcal{P} \times m \times \mathcal{S}$ for the target function $\vec{f} = [f_1, \cdots, f_m]$ (where $\mathcal{P}, \mathcal{S}$ are subsets of $\Sigma^*$ both containing $\lambda$), a common rank $R$, and task specific ranks $R_i$ for $i \in [m]$.

**Output:** WFAs $A_i$ approximating $f_i$ for each $i \in [d]$.

1: Compute the rank $R$ truncated SVD $\hat{\boldsymbol{\mathcal{H}}}_{(1)} \simeq \mathbf{U}\mathbf{D}\mathbf{V}^\top$.
2: Let $A = (\boldsymbol{\alpha}, \{\mathbf{A}^\sigma\}_{\sigma \in \Sigma}, \boldsymbol{\Omega})$ be the vv-WFA defined by

$$\boldsymbol{\alpha}^\top = \mathbf{U}_{\lambda,:}, \quad , \boldsymbol{\Omega} = \mathbf{U}^\top(\hat{\boldsymbol{\mathcal{H}}}_{:,:,\lambda}) \quad \text{and} \quad \mathbf{A}^\sigma = \mathbf{U}^\top \hat{\boldsymbol{\mathcal{H}}}^\sigma_{(1)}(\hat{\boldsymbol{\mathcal{H}}}_{(1)})^\dagger \mathbf{U} \ \text{ for each } \sigma \in \Sigma.$$

3: **for** $i = 1$ **to** $m$ **do**
4:     Compute the rank $R_i$ truncated SVD $\hat{\boldsymbol{\mathcal{H}}}_{:,i,:} \simeq \mathbf{U}_i \mathbf{D}_i \mathbf{V}_i^\top$.
5:     Let $A_i = \langle \mathbf{U}_i^\top \mathbf{U}\boldsymbol{\alpha}, \{\mathbf{U}_i^\top \mathbf{U}\mathbf{A}^\sigma \mathbf{U}^\top \mathbf{U}_i\}_{\sigma \in \Sigma}, \mathbf{U}_i^\top \mathbf{U}\boldsymbol{\Omega}_{:,i}\rangle$
6: **end for**
7: **return** $A_1, \cdots, A_m$.

---

## 4 Theoretical Analysis

**Computational complexity**. The computational cost of the classical spectral learning algorithm (SL) is in $\mathcal{O}\left(N + R|\mathcal{P}||\mathcal{S}| + R^2|\mathcal{P}||\Sigma|\right)$ where the first term corresponds to estimating the Hankel matrices from a sample of size $N$, the second one to the rank $R$ truncated SVD, and the third one to computing the transition matrices $\mathbf{A}^\sigma$. In comparison, the computational cost of MT-SL is in $\mathcal{O}\left(mN + (mR + \sum_i R_i)|\mathcal{P}||\mathcal{S}| + (mR^2 + \sum_i R_i^2)|\mathcal{P}||\Sigma|\right)$, showing that the increase in complexity is essentially linear in the number of tasks $m$.

**Robustness in subspace estimation**. In order to give some theoretical insights on the potential benefits of MT-SL, let us consider the simple case where the tasks are maximally related with common rank $R = R_1 = \cdots = R_m$. Let $\hat{\mathbf{H}}_1, \cdots, \hat{\mathbf{H}}_m \in \mathbb{R}^{\mathcal{P} \times \mathcal{S}}$ be the empirical Hankel matrices for the $m$ tasks and let $\mathbf{E}_i = \hat{\mathbf{H}}_i - \mathbf{H}_i$ be the error terms, where $\mathbf{H}_i$ is the true Hankel matrix for the $i$th task. Then the flattening $\hat{\mathbf{H}} = \hat{\boldsymbol{\mathcal{H}}}_{(1)} \in \mathbb{R}^{|\mathcal{P}| \times m|\mathcal{S}|}$ (resp. $\mathbf{H} = \boldsymbol{\mathcal{H}}_{(1)}$) can be obtained by stacking the matrices $\hat{\mathbf{H}}_i$ (resp. $\mathbf{H}_i$) along the horizontal axis. Consider the problem of learning the first task. One key step of both SL and MT-SL resides in estimating the left singular subspace of $\mathbf{H}_1$ and $\mathbf{H}$ respectively from their noisy estimates. When the tasks are maximally related, this space $\mathcal{U}$ is the same for $\mathbf{H}$ and $\mathbf{H}_1, \cdots, \mathbf{H}_m$ and we intuitively see that the benefits of MT-SL will stem from the fact that the SVD of $\hat{\mathbf{H}}$ should lead to a more accurate estimation of $\mathcal{U}$ than the one only relying on $\hat{\mathbf{H}}_1$. It is also intuitive to see that since the Hankel matrices $\hat{\mathbf{H}}_i$ have been stacked horizontally, the estimation of the right singular subspace might not benefit from performing SVD on $\hat{\mathbf{H}}$. However, classical results on singular subspace estimation (see e.g. [29, 20]) provide uniform bounds for both left and right singular subspaces (i.e. bounds on the maximum of the estimation errors for the left and right spaces). To circumvent this issue, we use a recent result on rate optimal asymmetric perturbation bounds for left and right singular spaces [9] to obtain the following theorem relating the ratio between the dimensions of a matrix to the quality of the subspace estimation provided by SVD (the proof can be found in the supplementary material).

**Theorem 5.** *Let* $\mathbf{M} \in \mathbb{R}^{d_1 \times d_2}$ *be of rank $R$ and let* $\hat{\mathbf{M}} = \mathbf{M} + \mathbf{E}$ *where* $\mathbf{E}$ *is a random noise term such that* $\frac{\text{vec}(\mathbf{E})}{\|\mathbf{E}\|_F}$ *follows a uniform distribution on the unit sphere in* $\mathbb{R}^{d_1 d_2}$*. Let* $\boldsymbol{\Pi}_U, \boldsymbol{\Pi}_{\hat{U}} \in \mathbb{R}^{d_1 \times d_1}$

*be the matrices of the orthogonal projections onto the space spanned by the top $R$ left singular vectors of $\mathbf{M}$ and $\hat{\mathbf{M}}$ respectively.*

*Let $\delta > 0$, let $\alpha = \mathfrak{s}_R(\mathbf{M})$ be the smallest non-zero singular value of $\mathbf{M}$ and suppose that $\|\mathbf{E}\|_F \le \alpha/2$. Then, with probability at least $1 - \delta$,*

$$\|\mathbf{\Pi}_U - \mathbf{\Pi}_{\hat{U}}\|_F \le 4 \left( \sqrt{\frac{(d_1 - R)R + 2\log(1/\delta)}{d_1 d_2}} \frac{\|\mathbf{E}\|_F}{\alpha} + \frac{\|\mathbf{E}\|_F^2}{\alpha^2} \right).$$

A few remarks on this theorem are in order. First, the Frobenius norm between the projection matrices measures the distance between the two subspaces (it is in fact proportional to the classical sin-theta distance between subspaces). Second, the assumption $\|\mathbf{E}\|_F \le \alpha/2$ corresponds to the magnitude of the noise being small compared to the magnitude of $\mathbf{M}$ (and in particular it implies $\frac{\|\mathbf{E}\|_F}{\alpha} < 1$); this is a reasonable and common assumption in subspace identification problems, see e.g. [30]. Lastly, as $d_2$ grows the first term in the upper bound becomes irrelevant and the error is dominated by the quadratic term, which decreases with $\|\mathbf{E}\|_F$ faster than classical results. Intuitively this tells us that there is a first regime where growing $d_2$ (i.e. adding more tasks) is beneficial, until the point where the quadratic term dominates (and where the bound becomes somehow independent of $d_2$).

Going back to the power of MT-SL to leverage information from related tasks, let $\mathbf{E} \in \mathbb{R}^{|\mathcal{P}| \times m|\mathcal{S}|}$ be the matrix obtained by stacking the noise matrices $\mathbf{E}_i$ along the horizontal axis. If we assume that the entries of the error terms $\mathbf{E}_i$ are i.i.d. from e.g. a normal distribution, we can apply the previous proposition to the left singular subspaces of $\hat{\mathcal{H}}_{(1)}$ and $\mathcal{H}_{(1)}$. One can check that in this case we have $\|\mathbf{E}\|_F^2 = \sum_{i=1}^m \|\mathbf{E}_i\|_F^2$ and $\alpha^2 = \mathfrak{s}_R(\mathbf{H})^2 \ge \sum_{i=1}^m \mathfrak{s}_R(\mathbf{H}_i)^2$ (since $R = R_1 = \cdots = R_m$ when tasks are maximally related). Thus, if the norms of the noise terms $\mathbf{E}_i$ are roughly the same, and so are the smallest non-zero singular values of the matrices $\mathbf{H}_i$, we get $\frac{\|\mathbf{E}\|_F}{\alpha} \le \mathcal{O}\left(\|\mathbf{E}_1\|_F/\mathfrak{s}_R(\mathbf{H}_1)\right)$. Hence, given enough tasks, the estimation error of the left singular subspace of $\mathbf{H}_1$ in the multitask setting (i.e. by performing SVD on $\hat{\mathcal{H}}_{(1)}$) is intuitively in $\mathcal{O}\left(\|\mathbf{E}_1\|_F^2/\mathfrak{s}_R(\mathbf{H}_1)^2\right)$ while it is only in $\mathcal{O}\left(\|\mathbf{E}_1\|_F/\mathfrak{s}_R(\mathbf{H}_1)\right)$ when relying solely on $\hat{\mathbf{H}}_1$, which shows the potential benefits of MT-SL. Indeed, as the amount of training data increases the error in the estimated matrices decreases, thus $T = \|\mathbf{E}_1\|_F/\mathfrak{s}_R(\mathbf{H}_1)$ goes to 0 and an error of order $\mathcal{O}\left(T^2\right)$ decays faster than one of order $\mathcal{O}\left(T\right)$.

## 5 Experiments

We evaluate the performance of the proposed multitask learning method (MT-SL) on both synthetic and real world data. We use two performance metrics: perplexity per character on a test set $T$, which is defined by $\text{perp}(h) = 2^{-\frac{1}{M} \sum_{x \in T} \log(h(x))}$ where $M$ is the number of symbols in the test set and $h$ is the hypothesis, and word error rate (WER) which measures the proportion of mis-predicted symbols averaged over all prefixes in the test set (when the most likely symbol is predicted). Both experiments are in a stochastic setting, i.e. the functions to be learned are probability distributions, and explore the regime where the learner has access to a small training sample drawn from the target task, while larger training samples are available for related tasks. We compare MT-SL with the classical spectral learning method (SL) for WFAs (note that SL has been extensively compared to EM and n-gram in the literature, see e.g. [4] and [5] and references therein). For both methods the prefix set $\mathcal{P}$ (resp. suffix set $\mathcal{S}$) is chosen by taking the $1,000$ most frequent prefixes (resp. suffixes) in the training data of the target task, and the values of the ranks are chosen using a validation set.

### 5.1 Synthetic Data

We first assess the validity of MT-SL on synthetic data. We randomly generated stochastic WFAs using the process used for the PAutomaC competition [27] with symbol sparsity $0.4$ and transition sparsity $0.15$, for an alphabet $\Sigma$ of size 10. We generated related WFAs[5] sharing a joint feature

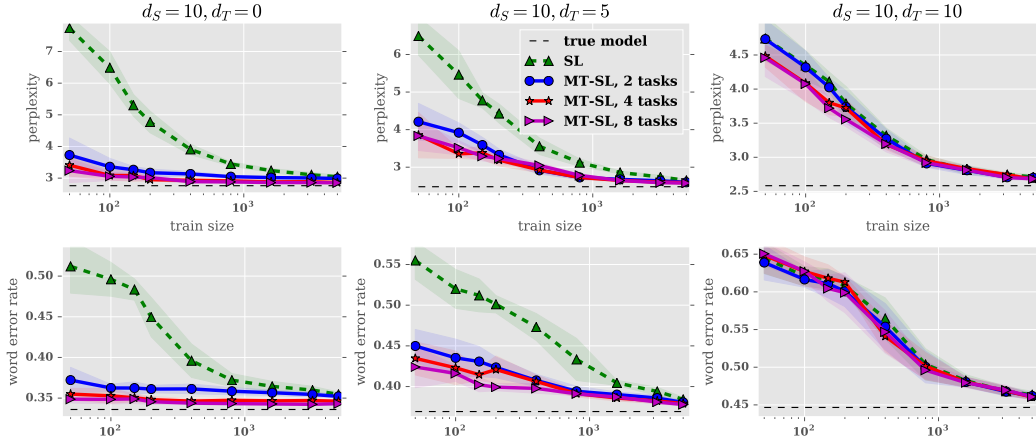

Figure 1: Comparison (on synthetic data) between the spectral learning algorithm (SL) and our multitask algorithm (MT-SL) for different numbers of tasks and different degrees of relatedness between the tasks: $d_S$ is the dimension of the space shared by all tasks and $d_T$ the one of the task-specific space (see text for details).

space of dimension $d_S = 10$ and each having a task specific feature space of dimension $d_T$, i.e. for $m$ tasks $f_1, \cdots, f_m$ each WFA computing $f_i$ has rank $d_S + d_T$ and the vv-WFA computing $\vec{f} = [f_1, \cdots, f_m]$ has rank $d_S + md_T$. We generated 3 sets of WFAs for different task specific dimensions $d_T = 0, 5, 10$. The learner had access to training samples of size $5,000$ drawn from each related tasks $f_2, \cdots, f_m$ and a training sample of sizes ranging from $50$ to $5,000$ drawn from the target task $f_1$. Results on a test set of size $1,000$ averaged over $10$ runs are reported in Figure 1.

For both evaluation measures, when the task specific dimension is small compared to the dimension of the joint feature space, i.e. $d_T = 0, 5$, MT-SL clearly outperforms SL that only relies on the target task data. Moreover, increasing the number of related tasks tends to improve the performances of MT-SL. However, when $d_S = d_T = 10$, MT-SL performs similarly in terms of perplexity and WER, showing that the multitask approach offers no benefits when the tasks are too loosely related. Additional experimental results for the case of totally unrelated tasks ($d_S = 0, d_T = 10$) as well as comparisons with MT-SL without the projection step (i.e. without lines 4-7 of Algorithm 1) are presented in the supplementary material.

## 5.2 Real Data

We evaluate MT-SL on 33 languages from the Universal Dependencies (UNIDEP) 1.4 treebank [24], using the 17-tag universal Part of Speech (PoS) tagset. This dataset contains sentences from various languages where each word is annotated with Google universal PoS tags [25], and thus can be seen as a collection of samples drawn from 33 distributions over strings on an alphabet of size 17. For each language, the available data is split between a training, a validation and a test set ($80\%, 10\%, 10\%$). For each language and for various sizes of training samples, we compare independently learning the target task with SL against using MT-SL to exploit training data from related tasks. We tested two ways of selecting the related tasks: (1) all other languages are used and (2) for each language we selected the 4 closest languages w.r.t. the distance between the subspaces spanned by the top 50 left singular vectors of their Hankel matrices[6].

We compare MT-SL against SL (using only the training data for the target task) and against a naive baseline where all data from different tasks are bagged together and used as a training set for SL (SL-bagging). We also include the results obtained using MT-SL without the projection step (MT-SL-noproj). We report the average relative improvement of MT-SL, SL-bagging and MT-SL-noproj w.r.t. SL over all languages in Table 1, e.g. for perplexity we report $100 \cdot (p_{sl} - p_{mt})/p_{sl}$ where $p_{sl}$ (resp. $p_{mt}$) is the perplexity obtained by SL (resp. MT-SL) on the test set. We see that the multitask approach leads to improved results for both metrics, that the benefits tend to be greater for small training sizes, and that restricting the number of auxiliary tasks is overall beneficial. To give a

Table 1: Average relative improvement with respect to single task spectral learning (SL) of the multitask approach (with and without the projection step: `MT-SL` and `MT-SL-noproj`) and the bagging baseline (`SL-bagging`) on the UNIDEP dataset.

(a) Perplexity average relative improvement (in %).

| Training size | 100 | 500 | 1000 | 5000 | all available data |
|---|---|---|---|---|---|
| | Related tasks: all other languages | | | | |
| `MT-SL` | 7.0744 ( $\pm_{7.76}$) | 3.6666 ( $\pm_{5.22}$) | 3.2879 ( $\pm_{5.17}$) | 3.4187 ( $\pm_{5.57}$) | 3.1574 ( $\pm_{5.48}$) |
| `MT-SL-noproj` | 2.9884 ( $\pm_{9.82}$) | 2.2469 ( $\pm_{7.49}$) | 0.8509 ( $\pm_{7.41}$) | 1.1658 ( $\pm_{6.59}$) | 0.6958 ( $\pm_{6.38}$) |
| `SL-bagging` | $-19.00$ ( $\pm_{29.1}$) | $-13.32$ ( $\pm_{22.4}$) | $-10.65$ ( $\pm_{19.7}$) | $-5.371$ ( $\pm_{14.6}$) | $-2.630$ ( $\pm_{13.0}$) |
| | Related tasks: 4 closest languages | | | | |
| `MT-SL` | 6.0069 ( $\pm_{6.76}$) | 4.3670 ( $\pm_{5.83}$) | 4.4049 ( $\pm_{5.50}$) | 2.9689 ( $\pm_{5.87}$) | 2.8229 ( $\pm_{5.90}$) |
| `MT-SL-noproj` | 4.5732 ( $\pm_{8.78}$) | 2.9421 ( $\pm_{7.83}$) | 2.4549 ( $\pm_{7.15}$) | 2.2166 ( $\pm_{6.82}$) | 2.1451 ( $\pm_{6.52}$) |
| `SL-bagging` | $-18.41$ ( $\pm_{28.4}$) | $-12.73$ ( $\pm_{22.0}$) | $-10.34$ ( $\pm_{20.1}$) | $-3.086$ ( $\pm_{12.7}$) | 0.1926 ( $\pm_{10.2}$) |

(b) WER average relative improvement (in %).

| Training size | 100 | 500 | 1000 | 5000 | all available data |
|---|---|---|---|---|---|
| | Related tasks: all other languages | | | | |
| `MT-SL` | 1.4919 ($\pm_{2.37}$) | 1.3786 ($\pm_{2.94}$) | 1.2281 ($\pm_{2.62}$) | 1.4964 ($\pm_{2.70}$) | 1.4932 ($\pm_{2.77}$) |
| `MT-SL-noproj` | $-5.763$ ($\pm_{6.82}$) | $-9.454$ ($\pm_{8.95}$) | $-9.197$ ($\pm_{7.25}$) | $-9.201$ ($\pm_{6.02}$) | $-9.600$ ($\pm_{5.55}$) |
| `SL-bagging` | $-3.067$ ($\pm_{10.8}$) | $-6.998$ ($\pm_{11.6}$) | $-7.788$ ($\pm_{9.88}$) | $-8.791$ ($\pm_{9.54}$) | $-8.611$ ($\pm_{9.74}$) |
| | Related tasks: 4 closest languages | | | | |
| `MT-SL` | 2.0883 ($\pm_{3.26}$) | 1.5175 ($\pm_{2.87}$) | 1.2961 ($\pm_{2.57}$) | 1.3080 ($\pm_{2.55}$) | 1.2160 ($\pm_{2.31}$) |
| `MT-SL-noproj` | $-4.139$ ($\pm_{5.10}$) | $-5.841$ ($\pm_{6.29}$) | $-5.399$ ($\pm_{6.26}$) | $-5.526$ ($\pm_{4.93}$) | $-5.556$ ($\pm_{4.90}$) |
| `SL-bagging` | 0.3372 ($\pm_{7.80}$) | $-3.045$ ($\pm_{8.12}$) | $-3.822$ ($\pm_{7.33}$) | $-4.350$ ($\pm_{6.90}$) | $-3.588$ ($\pm_{7.06}$) |

concrete example, on the Basque task with a training set of size 500, the WER was reduced from $\sim 76\%$ for SL to $\sim 70\%$ using all other languages as related tasks, and to $\sim 65\%$ using the 4 closest tasks (Finnish, Polish, Czech and Indonesian). Overall, both SL-bagging and MT-SL-noproj obtain worst performance than MT-SL (though MT-SL-noproj still outperforms SL in terms are perplexity while SL-bagging performs almost always worse than SL). Detailed results on all languages, along with the list of closest languages used for method (2), are reported in the supplementary material.

## 6  Conclusion

We introduced the novel model of vector-valued WFA that allowed us to define a notion of relatedness between recognizable functions and to design a multitask spectral learning algorithm for WFAs (MT-SL). The benefits of MT-SL have been theoretically motivated and showcased on both synthetic and real data experiments. In future works, we plan to apply MT-SL in the context of reinforcement learning and to identify other areas of machine learning where vv-WFAs could prove to be useful. It would also be interesting to investigate a weighted approach such as the one presented in [19] for classical spectral learning; this could prove useful to handle the case where the amount of available training data differs greatly between tasks.

**Acknowledgments**

G. Rabusseau acknowledges support of an IVADO postdoctoral fellowship. B. Balle completed this work while at Lancaster University. We thank NSERC and CIFAR for their financial support.

## Footnotes

*guillaume.rabusseau@mail.mcgill.ca

†pigem@amazon.co.uk

‡jpineau@cs.mcgill.ca

[4]Both definitions performed similarly in multitask experiments on the dataset used in Section 5.2, we thus chose multiple final vectors as a convention.

[5]More precisely, we first generate a probabilistic automaton (PA) $A_S = (\boldsymbol{\alpha}_S, \{\mathbf{A}_S^\sigma\}_{\sigma \in \Sigma}, \boldsymbol{\omega}_S)$ with $d_S$ states. Then, for each task $i = 1, \cdots, m$ we generate a second PA $A_T = (\boldsymbol{\alpha}_T, \{\mathbf{A}_T^\sigma\}_{\sigma \in \Sigma}, \boldsymbol{\omega}_T)$ with $d_T$ states and a random vector $\boldsymbol{\omega} \in [0,1]^{d_S + d_T}$. Both PAs are generated using the process described in [27]. The task $f_i$ is then obtained as the distribution computed by the stochastic WFA $\langle \boldsymbol{\alpha}_S \oplus \boldsymbol{\alpha}_T, \{\mathbf{A}_S^\sigma \oplus \mathbf{A}_T^\sigma\}_{\sigma \in \Sigma}, \tilde{\boldsymbol{\omega}} \rangle$ with $\tilde{\boldsymbol{\omega}} = \boldsymbol{\omega}/Z$ where the constant $Z$ is chosen such that $\sum_{x \in \Sigma^*} f_i(x) = 1$.

[6]The common basis $(\mathcal{P}, \mathcal{S})$ for these Hankel matrices is chosen by taking the union of the 100 most frequent prefixes and suffixes in each training sample.

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
