[Supplementary Material]

# Multitask Spectral Learning of Weighted Automata (Supplementary Material)

**Guillaume Rabusseau** [*]
McGill University

**Borja Balle** [†]
Amazon Research Cambridge

**Joelle Pineau**[‡]
McGill University

# 1 Proofs

## 1.1 Proof of Theorem 3

**Theorem.** *Let $\vec{f} : \Sigma^* \to \mathbb{R}^d$ and let $\mathcal{H}$ be the corresponding Hankel tensor. Then* $\mathrm{rank}(f) = \mathrm{rank}(\mathcal{H}_{(1)})$.

*Proof.* We first show that $\mathrm{rank}(\vec{f}) \geq \mathrm{rank}(\mathcal{H}_{(1)})$. Let $A = (\boldsymbol{\alpha}, \{\mathbf{A}^\sigma\}_{\sigma \in \Sigma}, \boldsymbol{\Omega})$ be a vv-WFA with $n$ states computing $\vec{f}$ and let $\mathbf{P} \in \mathbb{R}^{\Sigma^* \times n}$ and $\mathcal{S} \in \mathbb{R}^{n \times d \times \Sigma^*}$ be defined by

$$\mathbf{P}_{u,:} = \boldsymbol{\alpha}^\top \mathbf{A}^u \ \text{ and } \ \mathcal{S}_{:,:,v} = \mathbf{A}^v \boldsymbol{\Omega}.$$

It is easy to check that $\mathcal{H} = \mathcal{S} \times_1 \mathbf{P}$ which implies $\mathcal{H}_{(1)} = \mathbf{P}\mathcal{S}_{(1)}$ and thus $\mathrm{rank}(\mathcal{H}_{(1)}) \leq n$.

For the converse, we first define the notion of *residual functions* of $\vec{f}$: for any $x \in \Sigma^*$ the residual $\overline{x} : \Sigma^* \to \mathbb{R}^d$ is the function defined by $\overline{x}(u) = \vec{f}(xu)$ for any $u \in \Sigma^*$. Let $V = \{\overline{x} \ : \ x \in \Sigma^*\} \subset (\mathbb{R}^d)^{\Sigma^*}$ be the space of residual functions of $f$. Suppose that $\mathrm{rank}(\mathcal{H}_{(1)}) = n$. Since each residual $\overline{x}$ can be identified with the row vector $(\mathcal{H}_{(1)})_{x,:}$, the dimension of $V$ is equal to $n$. Thus there exist $n$ words $e_1, \cdots, e_n \in \Sigma^*$ such that $(\overline{e_1}, \cdots, \overline{e_n})$ is a basis of $V$. Expressing $\overline{\lambda}$ and $\overline{e_i\sigma}$ for each $i \in [n]$, $\sigma \in \Sigma$ in this basis, we know that there exist $\boldsymbol{\alpha} \in \mathbb{R}^n$ and $\mathbf{A}^\sigma \in \mathbb{R}^{n \times n}$ for each $\sigma$ such that

$$\overline{\lambda} = \sum_i \boldsymbol{\alpha}_i \overline{e_i} \ \text{ and } \ \overline{e_i\sigma} = \sum_j \mathbf{A}^\sigma_{i,j} \overline{e_j}.$$

We now show by induction on $|x|$ that $\overline{e_ix} = \sum_j \mathbf{A}^x_{i,j} \overline{e_j}$ for any non-empty string $x \in \Sigma^*$. The case $x = \sigma \in \Sigma$ is immediate by definition of $\mathbf{A}^\sigma$. Let $x, y$ be two non-empty words, for any $u \in \Sigma^*$ and any $i \in [n]$ we get

$$\overline{e_ixy}(u) = \vec{f}(e_ixyu) = \overline{e_ix}(yu)$$
$$= \sum_j \mathbf{A}^x_{i,j} \overline{e_j}(yu) = \sum_j \mathbf{A}^x_{i,j} \vec{f}(e_jyu) = \sum_j \mathbf{A}^x_{i,j} \overline{e_jy}(u)$$
$$= \sum_j \mathbf{A}^x_{i,j} \sum_k \mathbf{A}^y_{j,k} \overline{e_k}(u) = \sum_k \mathbf{A}^{xy}_{i,k} \overline{e_k}(u)$$

using the induction hypothesis twice. To conclude the proof, let $\boldsymbol{\Omega} \in \mathbb{R}^{n \times d}$ be the matrix with rows $\overline{e_i}(\lambda)$ for $i \in [n]$. For any $x \in \Sigma^*$ we have

$$\vec{f}(x) = \overline{\lambda}(x) = \sum_i \boldsymbol{\alpha}_i \overline{e_i}(x) = \sum_i \boldsymbol{\alpha}_i \overline{e_ix}(\lambda)$$
$$= \sum_i \boldsymbol{\alpha}_i \sum_j \mathbf{A}^x_{i,j} \overline{e_j}(\lambda) = \boldsymbol{\alpha}^\top \mathbf{A}^x \boldsymbol{\Omega},$$

[*]guillaume.rabusseau@mail.mcgill.ca

[†]pigem@amazon.co.uk

[‡]jpineau@cs.mcgill.ca

showing that the vv-WFA $(\boldsymbol{\alpha}, \{\mathbf{A}^\sigma\}_{\sigma \in \Sigma}, \boldsymbol{\Omega})$ computes $\vec{f}$ and consequently that $\mathrm{rank}(\vec{f}) \leq n = \mathrm{rank}(\boldsymbol{\mathcal{H}}_{(1)})$. $\qquad\square$

## 1.2 Proof of Corollary 4

**Corollary.** *Let $\vec{f} : \Sigma^* \to \mathbb{R}^d$ be a recognizable function with rank n, let $\boldsymbol{\mathcal{H}} \in \mathbb{R}^{\Sigma^* \times d \times \Sigma^*}$ be its Hankel tensor, and for each $\sigma \in \Sigma$ let $\boldsymbol{\mathcal{H}}^\sigma \in \mathbb{R}^{\Sigma^* \times d \times \Sigma^*}$ be defined by $\boldsymbol{\mathcal{H}}^\sigma_{u,:,v} = f(u\sigma v)$ for all $u, v \in \Sigma^*$.*

*Then, for any $\mathbf{P} \in \mathbb{R}^{\Sigma^* \times n}$ and $\boldsymbol{\mathcal{S}} \in \mathbb{R}^{n \times d \times \Sigma^*}$ such that $\boldsymbol{\mathcal{H}} = \boldsymbol{\mathcal{S}} \times_1 \mathbf{P}$, the vv-WFA $A = (\boldsymbol{\alpha}, \{\mathbf{A}^\sigma\}_{\sigma \in \Sigma}, \boldsymbol{\Omega})$ defined by $\boldsymbol{\alpha}^\top = \mathbf{P}_{\lambda,:}$, $\boldsymbol{\Omega} = \boldsymbol{\mathcal{S}}_{:,:,\lambda}$, and $\mathbf{A}^\sigma = \mathbf{P}^\dagger \boldsymbol{\mathcal{H}}^\sigma_{(1)} (\boldsymbol{\mathcal{S}}_{(1)})^\dagger$ is a minimal vv-WFA computing $\vec{f}$.*

*Proof.* Let $\hat{A} = (\hat{\boldsymbol{\alpha}}^\top, \{\hat{\mathbf{A}}^\sigma\}_{\sigma \in \Sigma}, \hat{\boldsymbol{\Omega}})$ be a minimal vv-WFA computing $\vec{f}$ and let $\hat{\mathbf{P}} \in \mathbb{R}^{\Sigma^* \times n}$ and $\hat{\boldsymbol{\mathcal{S}}} \in \mathbb{R}^{n \times d \times \Sigma^*}$ be defined by

$$\hat{\mathbf{P}}_{u,:} = \boldsymbol{\alpha}^\top \hat{\mathbf{A}}^u \ \text{ and } \ \hat{\boldsymbol{\mathcal{S}}}_{:,:,v} = \hat{\mathbf{A}}^v \boldsymbol{\Omega}, \ \ u, v \in \Sigma^*,$$

hence $\boldsymbol{\mathcal{H}} = \hat{\boldsymbol{\mathcal{S}}} \times_1 \hat{\mathbf{P}}$ and, equivalently, $\boldsymbol{\mathcal{H}}_{(1)} = \hat{\mathbf{P}} \hat{\boldsymbol{\mathcal{S}}}_{(1)}$. We will show that $\boldsymbol{\alpha}^\top = \hat{\boldsymbol{\alpha}}^\top \mathbf{M}^{-1}, \boldsymbol{\Omega} = \mathbf{M} \hat{\boldsymbol{\Omega}}$ and $\mathbf{A}^\sigma = \mathbf{M} \hat{\mathbf{A}}^\sigma \mathbf{M}^{-1}$ for each $\sigma \in \Sigma$ where $\mathbf{M} = \mathbf{P}^\dagger \hat{\mathbf{P}}$, which will imply $\vec{f}_A = \vec{f}_{\hat{A}} = \vec{f}$.

To simplify the notations, let $\mathbf{H} = \boldsymbol{\mathcal{H}}_{(1)}, \mathbf{S} = \boldsymbol{\mathcal{S}}_{(1)}, \hat{\mathbf{S}} = \hat{\boldsymbol{\mathcal{S}}}_{(1)}$, and $\mathbf{H}^\sigma = (\boldsymbol{\mathcal{H}}^\sigma)_{(1)}$ for each $\sigma \in \Sigma$. First observe that since $\mathbf{P}^\dagger \hat{\mathbf{P}} \hat{\mathbf{S}} \mathbf{S}^\dagger = \mathbf{P}^\dagger \mathbf{H} \mathbf{S}^\dagger = \mathbf{I}$, the matrix $\mathbf{M}$ is invertible with $\mathbf{M}^{-1} = \hat{\mathbf{S}} \mathbf{S}^\dagger$. Using the identities $\mathbf{H}^\sigma = \hat{\mathbf{P}} \hat{\mathbf{A}}^\sigma \hat{\mathbf{S}}, \mathbf{H}_{\lambda,:} = \hat{\boldsymbol{\alpha}}^\top \hat{\mathbf{S}}, \mathbf{P}^\dagger \boldsymbol{\mathcal{H}}_{:,:,\lambda} = \boldsymbol{\mathcal{S}}_{:,:,\lambda}$, and $\boldsymbol{\mathcal{H}}_{:,:,\lambda} = \hat{\mathbf{P}} \hat{\boldsymbol{\Omega}}$, we then get

$$\mathbf{A}^\sigma = \mathbf{P}^\dagger \mathbf{H}^\sigma \mathbf{S}^\dagger = \mathbf{P}^\dagger \hat{\mathbf{P}} \hat{\mathbf{A}}^\sigma \hat{\mathbf{S}} \mathbf{S}^\dagger = \mathbf{M} \hat{\mathbf{A}}^\sigma \mathbf{M}^{-1},$$

$$\boldsymbol{\alpha}^\top = \mathbf{P}_{\lambda,:} = \mathbf{H}_{\lambda,:} \mathbf{S}^\dagger = \hat{\boldsymbol{\alpha}}^\top \hat{\mathbf{S}} \mathbf{S}^\dagger = \hat{\boldsymbol{\alpha}}^\top \mathbf{M}^{-1}, \text{ and}$$

$$\boldsymbol{\Omega} = \boldsymbol{\mathcal{S}}_{:,:,\lambda} = \mathbf{P}^\dagger \boldsymbol{\mathcal{H}}_{:,:,\lambda} = \mathbf{P}^\dagger \hat{\mathbf{P}} \hat{\boldsymbol{\Omega}} = \mathbf{M} \hat{\boldsymbol{\Omega}}. \qquad\square$$

## 1.3 Proof of Proposition 1

**Proposition.** *Let $\vec{f} : \Sigma^* \to \mathbb{R}^d$ be a function computed by a vv-WFA $A = (\boldsymbol{\alpha}, \{\mathbf{A}^\sigma\}_{\sigma \in \Sigma}, \boldsymbol{\Omega})$ with n states and let $\mathcal{P}, \mathcal{S} \subseteq \Sigma^*$ be a complete basis for $\vec{f}$. For any $i \in [d]$, let $f_i : \Sigma^* \to \mathbb{R}$ be defined by $f_i(x) = \vec{f}(x)_i$ for all $x \in \Sigma^*$ and let $n_i$ denote the rank of $f_i$.*

*Let $\mathbf{P} \in \mathbb{R}^{\mathcal{P} \times n}$ be defined by $\mathbf{P}_{x,:} = \boldsymbol{\alpha}^\top \mathbf{A}^x$ for all $x \in \mathcal{P}$ and, for $i \in [d]$, let $\mathbf{H}_i = \mathbf{U}_i \mathbf{D}_i \mathbf{V}_i^\top$ be the thin SVD of $\mathbf{H}_i$ (i.e. $\mathbf{D}_i \in \mathbb{R}^{n_i \times n_i}$) where $\mathbf{H}_i \in \mathbb{R}^{\mathcal{P} \times \mathcal{S}}$ is the hankel matrix of $f_i$.*

*Then, for any $i \in [d]$, the WFA $A_i = \langle \boldsymbol{\alpha}_i, \{\mathbf{A}_i^\sigma\}_{\sigma \in \Sigma}\}, \boldsymbol{\omega}_i \rangle$ defined by*

$$\boldsymbol{\alpha}_i^\top = \boldsymbol{\alpha}^\top \mathbf{P}^\dagger \mathbf{U}_i, \boldsymbol{\omega}_i = \mathbf{U}_i^\top \mathbf{P} \boldsymbol{\Omega}_{:,i} \text{ and } \mathbf{A}_i^\sigma = \mathbf{U}_i^\top \mathbf{P} \mathbf{A}^\sigma \mathbf{P}^\dagger \mathbf{U}_i \text{ for each } \sigma \in \Sigma,$$

*is a minimal WFA computing $f_i$.*

*Proof.* For each $i \in [d]$, let $\mathbf{S}_i \in \mathbb{R}^{n \times \mathcal{S}}$ be defined by $(\mathbf{S}_i)_{:,x} = \mathbf{A}^x \boldsymbol{\Omega}_{:,i}$ and consider the $|\mathcal{P}| \times d|\mathcal{S}|$ block matrices $\mathbf{H} = \begin{bmatrix} \mathbf{H}_1, & \cdots & ,\mathbf{H}_d \end{bmatrix}, \mathbf{H}^\sigma = \begin{bmatrix} \mathbf{H}_1^\sigma, & \cdots & ,\mathbf{H}_d^\sigma \end{bmatrix}$ for each $\sigma \in \Sigma$, and $\mathbf{S} = \begin{bmatrix} \mathbf{S}_1, & \cdots & ,\mathbf{S}_d \end{bmatrix}$.

We show the result for $i = 1$. First, it follows from applying Corollary 2 to the factorization $\mathbf{H}_1 = \mathbf{U}_1(\mathbf{D}_1 \mathbf{V}_1^\top)$ that the WFA $\hat{A} = \langle \hat{\boldsymbol{\alpha}}, \{\hat{\mathbf{A}}^\sigma\}_{\sigma \in \Sigma}, \hat{\boldsymbol{\omega}} \rangle$ defined by

$$\hat{\boldsymbol{\alpha}}^\top = (\mathbf{U}_1)_{\lambda,:}, \hat{\boldsymbol{\omega}} = (\mathbf{D}\mathbf{V}_1^\top)_{:,\lambda} \text{ and } \hat{\mathbf{A}}^\sigma = \mathbf{U}_1^\top \mathbf{H}_1^\sigma \mathbf{V}_1 \mathbf{D}_1^{-1} \text{ for each } \sigma \in \Sigma$$

is a minimal WFA computing $f_1$. We will show that the WFA $A_1$ is exactly $\hat{A}$.

Let $\sigma \in \Sigma$. We start by showing that $\mathbf{A}_1^\sigma = \hat{\mathbf{A}}^\sigma$. It is easy to check that $\mathbf{H} = \mathbf{P}\mathbf{S}$ and $\mathbf{H}^\sigma = \mathbf{P}\mathbf{A}^\sigma \mathbf{S}$. Furthermore, since $\mathbf{H}^\sigma = \mathbf{H}^\sigma \mathbf{S}^\dagger \mathbf{S}$ we have $\mathbf{H}_1^\sigma = \mathbf{H}^\sigma \mathbf{S}^\dagger \mathbf{S}_1$, which implies $\mathbf{A}^\sigma \mathbf{S}_1 = \mathbf{P}^\dagger \mathbf{H}^\sigma \mathbf{S}^\dagger \mathbf{S}_1 = $

$\mathbf{P}^\dagger \mathbf{H}_1^\sigma$. It then follows that

$$
\begin{aligned}
\mathbf{A}_1^\sigma &= \mathbf{U}_1^\top \mathbf{P} \mathbf{A}^\sigma \mathbf{P}^\dagger \mathbf{U}_1 \\
&= \mathbf{U}_1^\top \mathbf{P} \mathbf{A}^\sigma \mathbf{P}^\dagger \mathbf{U}_1 (\mathbf{D}_1 \mathbf{V}_1^\top \mathbf{V}_1 \mathbf{D}_1^{-1}) \\
&= \mathbf{U}_1^\top \mathbf{P} \mathbf{A}^\sigma \mathbf{P}^\dagger \mathbf{H}_1 \mathbf{V}_1 \mathbf{D}_1^{-1} \\
&= \mathbf{U}_1^\top \mathbf{P} \mathbf{A}^\sigma \mathbf{S}_1 \mathbf{V}_1 \mathbf{D}_1^{-1} \\
&= \mathbf{U}_1^\top \mathbf{P} \mathbf{P}^\dagger \mathbf{H}_1^\sigma \mathbf{V}_1 \mathbf{D}_1^{-1} \\
&= \mathbf{U}_1^\top \mathbf{H}_1^\sigma \mathbf{V}_1 \mathbf{D}_1^{-1} = \hat{\mathbf{A}}^\sigma
\end{aligned}
$$

where we also used the fact that $\mathbf{P}\mathbf{P}^\dagger \mathbf{H}_1^\sigma = \mathbf{H}_1^\sigma$ and $\mathbf{P}^\dagger \mathbf{H}_1 = \mathbf{S}_1$. Now since the column space of $\mathbf{U}_1$ is contained in the column space of $\mathbf{P}$, we have $\mathbf{U}_1^\top \mathbf{P}\mathbf{P}^\dagger = \mathbf{U}_1^\top$ (and similarly $\mathbf{P}\mathbf{P}^\dagger \mathbf{U}_1 = \mathbf{U}_1$). Using the this fact and observing that $\boldsymbol{\alpha}^\top = \mathbf{H}_{\lambda,:} \mathbf{S}^\dagger$ and $\boldsymbol{\Omega} = \mathbf{P}^\dagger(\boldsymbol{\mathcal{H}}_{:,:,\lambda})$ we get

$$
\boldsymbol{\alpha}_1^\top = \boldsymbol{\alpha}^\top \mathbf{P}^\dagger \mathbf{U}_1 = \mathbf{H}_{\lambda,:} \mathbf{S}^\dagger \mathbf{P}^\dagger \mathbf{U}_1 = \mathbf{P}_{\lambda,:} \mathbf{P}^\dagger \mathbf{U}_1 = (\mathbf{U}_1)_{\lambda,:} = \hat{\boldsymbol{\alpha}}^\top
$$

and

$$
\boldsymbol{\omega}_1 = \mathbf{U}_1^\top \mathbf{P} \boldsymbol{\Omega}_{:,1} = \mathbf{U}_1^\top \mathbf{P}\mathbf{P}^\dagger(\boldsymbol{\mathcal{H}}_{:,1,\lambda}) = \mathbf{U}_1^\top (\mathbf{H}_1)_{:,\lambda} = \hat{\boldsymbol{\omega}}
$$

which concludes the proof. $\qquad \square$

## 1.4 Proof of Theorem 5

**Theorem.** *Let* $\mathbf{M} \in \mathbb{R}^{d_1 \times d_2}$ *be of rank* $R$ *and let* $\hat{\mathbf{M}} = \mathbf{M} + \mathbf{E}$ *where* $\mathbf{E}$ *is a random noise term such that* $\frac{\mathrm{vec}(\mathbf{E})}{\|\mathbf{E}\|_F}$ *follows a uniform distribution on the unit sphere in* $\mathbb{R}^{d_1 d_2}$. *Let* $\boldsymbol{\Pi}_U, \boldsymbol{\Pi}_{\hat{U}} \in \mathbb{R}^{d_1 \times d_1}$ *be the matrices of the orthogonal projections onto the space spanned by the top* $R$ *left singular vectors of* $\mathbf{M}$ *and* $\hat{\mathbf{M}}$ *respectively.*

*Let* $\delta > 0$, *let* $\alpha = \mathfrak{s}_R(\mathbf{M})$ *be the smallest non-zero singular value of* $\mathbf{M}$ *and suppose that* $\|\mathbf{E}\|_F \leq \alpha/2$. *Then, with probability at least* $1 - \delta$,

$$
\|\boldsymbol{\Pi}_U - \boldsymbol{\Pi}_{\hat{U}}\|_F \leq 4 \left( \sqrt{\frac{(d_1 - R)R + 2\log(1/\delta)}{d_1 d_2}} \frac{\|\mathbf{E}\|_F}{\alpha} + \frac{\|\mathbf{E}\|_F^2}{\alpha^2} \right).
$$

Let $\boldsymbol{\Pi}_{U_\perp} = \mathbf{I} - \boldsymbol{\Pi}_U$ and $\boldsymbol{\Pi}_{V_\perp} = \mathbf{I} - \boldsymbol{\Pi}_V$. Then, under the assumption $\|\mathbf{E}\|_F \leq \alpha/2$, it follows from Theorem 1 in [**?** ] that

$$
\|\boldsymbol{\Pi}_U - \boldsymbol{\Pi}_{\hat{U}}\|_F \leq \frac{2\sqrt{2}}{\alpha} \left( \|\boldsymbol{\Pi}_{U_\perp} \mathbf{E} \boldsymbol{\Pi}_V\|_F + \frac{\|\boldsymbol{\Pi}_{U_\perp} \mathbf{E} \boldsymbol{\Pi}_{V_\perp}\|_F \cdot \|\boldsymbol{\Pi}_U \mathbf{E} \boldsymbol{\Pi}_{V_\perp}\|_F}{\alpha} \right).
$$

The second term of the sum can be bounded using the fact that both $\|\boldsymbol{\Pi}_{U_\perp} \mathbf{E} \boldsymbol{\Pi}_{V_\perp}\|_F$ and $\|\boldsymbol{\Pi}_U \mathbf{E} \boldsymbol{\Pi}_{V_\perp}\|_F$ are bounded by $\|\mathbf{E}\|_F$. Indeed we have e.g.

$$
\|\boldsymbol{\Pi}_{U_\perp} \mathbf{E} \boldsymbol{\Pi}_{V_\perp}\|_F = \|(\boldsymbol{\Pi}_{V_\perp} \otimes \boldsymbol{\Pi}_{U_\perp})\mathrm{vec}(\mathbf{E})\|_F \leq \|\mathrm{vec}(\mathbf{E})\|_F = \|\mathbf{E}\|_F
$$

since $\boldsymbol{\Pi}_{V_\perp} \otimes \boldsymbol{\Pi}_{U_\perp}$ is the matrix of an orthogonal projection. To bound the first term, we use the following lemma showing that the norm of a $d$-dimensional random vector $\mathbf{v}$ projected onto a fixed subspace of dimension $k$ will be concentrated around $\sqrt{k/d}\|\mathbf{v}\|$.

**Lemma 1.** *Let* $\boldsymbol{\Pi} \in \mathbb{R}^{d \times d}$ *be a rank* $k$ *projection matrix and let* $\mathbf{v} \in \mathbb{R}^d$ *be a random variable such that* $\frac{\mathbf{v}}{\|\mathbf{v}\|}$ *follows a uniform distribution on the unit sphere in* $\mathbb{R}^d$. *Then, for any* $\delta > 0$,

$$
\mathbb{P}\left[ \|\boldsymbol{\Pi}\mathbf{v}\|_2^2 > 2\frac{k + 2\log(1/\delta)}{d} \|\mathbf{v}\|_2^2 \right] \leq \delta.
$$

*Proof.* This directly comes form the following classical result (see e.g. Lemma 2.4 in [**?** ]): if $\mathbf{x}$ is a random unit vector drawn uniformly from the unit sphere we have for any $\beta > 1$

$$
\mathbb{P}\left[ \|\boldsymbol{\Pi}\mathbf{x}\|_2^2 \leq \beta \frac{k}{d} \right] \leq \exp\left\{ \frac{k}{2}(1 - \beta + \log\beta) \right\}.
$$

Using the inequality $\log\beta \leq \beta/2$, the right term can be upper bounded by $\exp(k/2(1 - \beta/2))$, and by setting this upper bound equal to $\delta$ we get $\beta = 2(1 + 2\log(1/\delta)/k)$ which leads to the result. $\qquad \square$

Applying this lemma to $\|\mathbf{\Pi}_{U_\perp}\mathbf{E}\mathbf{\Pi}_V\|_F = \|(\mathbf{\Pi}_V \otimes \mathbf{\Pi}_{U_\perp})\text{vec}(\mathbf{E})\|_2$ by observing that $\mathbf{\Pi}_V \otimes \mathbf{\Pi}_{U_\perp}$ is a $d_1 d_2 \times d_1 d_2$ projection matrix of rank $R(d_1 - R)$, we get that $\|\mathbf{\Pi}_{U_\perp}\mathbf{E}\mathbf{\Pi}_V\|_F \leq \sqrt{2\frac{(d_1-R)R+2\log(1/\delta)}{d_1 d_2}}\|\mathbf{E}\|_F$ with probability at least $1 - \delta$ which concludes the proof.

## 2 Additional Experiments on Synthetic Data

In Figure 1, we present additional experimental results on the synthetic datasets: we added the case of totally unrelated tasks ($d_S = 0, d_T = 10$) and we included the performances of MT-SL-noproj in all plots. These additional results show that the projection step of MT-SL is crucial when the tasks are loosely related or totally unrelated, and that the results for the case of totally unrelated tasks are similar to the ones obtained when the tasks are only loosely related ($d_S = 10, d_T = 10$).

## 3 Detailed Results for Experiments on Real Data

The perplexity and WER on the test sets for all languages are reported in Table 1 when MT-SL is used with all other languages as related tasks, and in Table 2 when only the 4 closest languages are used. The list of the closest languages used for each task can be found in Table 3.

Figure 1: Additional experimental results on the synthetic datasets.

Table 1: Detailed experimental results on the UNIDEP dataset when all other languages are used as related tasks.

| Language | Training size | Perplexity | | | | Word Error Rate | | | |
|---|---|---|---|---|---|---|---|---|---|
| | | SL | SL bagging | MT-SL | MT-SL no proj. | SL | SL bagging | MT-SL | MT-SL no proj. |
| Ancient Greek | 100 | 4.053 | 4.997 | 4.030 | 4.266 | 82.445 | **76.293** | 82.445 | 79.883 |
| | 500 | 4.194 | 5.000 | 4.195 | 4.233 | 81.572 | 76.211 | **74.438** | 78.768 |
| | 1000 | 4.203 | 4.995 | 4.192 | 4.246 | 77.696 | 76.185 | **75.020** | 77.601 |
| | all | 4.582 | 4.925 | 4.564 | 4.676 | 75.596 | 75.875 | 75.804 | 79.972 |
| Arabic | 100 | 2.316 | 2.347 | 2.258 | 2.291 | 77.306 | 77.264 | 77.161 | 82.452 |
| | 500 | 2.298 | 2.347 | 2.298 | 2.298 | 68.804 | 77.285 | 68.804 | 73.954 |
| | 1000 | 2.315 | 2.347 | 2.311 | 2.311 | 68.066 | 77.288 | 68.066 | 74.109 |
| | all | 2.306 | 2.336 | 2.338 | 2.338 | 66.595 | 77.578 | 66.595 | 73.668 |
| Basque | 100 | 5.932 | 9.370 | 5.932 | 6.842 | 75.956 | 77.305 | **73.515** | 75.796 |
| | 500 | 6.057 | 9.368 | **5.950** | 7.009 | 76.300 | 77.240 | **70.351** | 75.223 |
| | 1000 | 6.261 | 9.364 | 6.261 | 7.369 | 77.129 | 77.290 | **68.666** | 74.329 |
| | all | 6.760 | 9.276 | 6.760 | 7.640 | 75.803 | 76.602 | **68.192** | 75.754 |
| Bulgarian | 100 | 5.103 | 6.659 | **4.722** | 5.613 | 74.647 | **67.490** | 73.525 | 73.525 |
| | 500 | 5.475 | 6.649 | 5.475 | 5.669 | 68.576 | 67.644 | **65.312** | 73.329 |
| | 1000 | 5.649 | 6.617 | 5.649 | 5.888 | 66.059 | 67.875 | **64.279** | 76.350 |
| | all | 6.162 | 6.485 | 6.162 | 6.380 | 66.018 | 70.736 | **62.196** | 73.389 |
| Croatian | 100 | 5.510 | 4.946 | 4.870 | 4.870 | 75.468 | 76.694 | 75.653 | 79.445 |
| | 500 | 5.336 | **4.946** | 5.167 | 5.167 | 77.503 | 76.717 | 77.757 | 79.098 |
| | 1000 | 5.454 | **4.947** | 5.257 | 5.257 | 77.526 | 76.532 | 77.757 | **75.329** |
| | all | 5.285 | **4.938** | 5.260 | 5.260 | 77.919 | 76.624 | 76.000 | 77.942 |
| Czech | 100 | 4.292 | 5.319 | 4.292 | 4.357 | 77.897 | **75.868** | 77.262 | 77.760 |
| | 500 | 4.482 | 5.316 | 4.444 | 4.501 | 77.257 | 75.899 | **71.326** | 75.608 |
| | 1000 | 4.530 | 5.310 | 4.530 | 4.533 | 75.254 | 75.887 | 74.471 | 74.623 |
| | all | 5.091 | 5.141 | 5.091 | 5.125 | 73.849 | 73.382 | **71.325** | 75.827 |
| Danish | 100 | 4.977 | 5.016 | **4.838** | 5.011 | 78.891 | **75.137** | 79.133 | 80.986 |
| | 500 | 5.033 | 5.054 | 5.034 | 5.034 | 71.737 | 75.185 | 71.737 | 81.228 |
| | 1000 | 5.189 | 5.062 | 5.016 | 5.016 | 71.737 | 75.314 | 71.737 | 79.842 |
| | all | 5.363 | **4.999** | 5.176 | 5.176 | 70.674 | 74.460 | 70.674 | 78.714 |
| Dutch | 100 | 6.456 | 6.592 | **5.603** | 6.302 | 77.525 | 75.649 | 75.113 | 78.094 |
| | 500 | 7.679 | 6.556 | 7.694 | **6.402** | 75.498 | 75.314 | 75.498 | 79.434 |
| | 1000 | 7.517 | **6.563** | 7.327 | 7.438 | 73.874 | 75.264 | 73.874 | 79.434 |
| | all | 8.025 | **6.524** | 8.201 | 8.201 | 72.785 | 74.711 | 72.785 | 76.101 |
| English | 100 | 5.203 | 6.862 | **4.874** | 5.086 | 73.853 | 74.397 | 73.853 | 77.573 |
| | 500 | **5.620** | 6.820 | 5.774 | 5.722 | 71.729 | 74.508 | 71.777 | 79.741 |
| | 1000 | 5.781 | 6.805 | 5.748 | 5.748 | 72.716 | 74.559 | **69.672** | 72.925 |
| | all | 6.464 | 6.572 | 6.464 | 6.595 | 67.626 | 74.143 | 67.626 | 74.213 |
| Estonian | 100 | 5.301 | 9.339 | **4.534** | 5.219 | 50.506 | 71.389 | 50.506 | 57.866 |
| | 500 | 6.108 | 8.776 | 5.625 | **5.341** | **49.586** | 72.493 | 50.874 | 68.169 |
| | 1000 | 6.605 | 8.557 | **6.254** | 7.443 | 50.138 | 68.721 | 50.138 | 57.222 |
| | all | 6.653 | 8.522 | **5.706** | 7.221 | 50.046 | 68.629 | 50.414 | 58.786 |
| Finnish | 100 | 5.598 | 9.418 | **5.153** | 5.153 | 68.774 | 70.140 | 68.774 | 72.302 |
| | 500 | 5.964 | 9.370 | **5.712** | 5.970 | 67.586 | 70.062 | 66.701 | 74.047 |
| | 1000 | 6.231 | 9.290 | **5.844** | 6.008 | 66.366 | 70.012 | 65.724 | 74.778 |
| | all | 7.709 | 8.580 | **7.420** | 7.642 | 63.811 | 70.893 | 62.848 | 66.612 |
| French | 100 | 3.744 | 3.843 | **3.273** | 3.493 | 69.245 | **66.115** | 68.658 | 70.831 |
| | 500 | 3.611 | 3.855 | 3.611 | 3.720 | 63.669 | 66.143 | 63.600 | 71.911 |
| | 1000 | 3.735 | 3.843 | 3.657 | 3.657 | 59.787 | 66.170 | 60.662 | 74.194 |
| | all | 3.823 | 3.830 | 3.823 | 3.875 | 59.732 | 65.377 | 59.732 | 69.615 |
| German | 100 | 5.756 | 5.825 | **5.234** | 5.234 | 78.498 | **74.769** | 78.458 | 80.864 |
| | 500 | 5.960 | 5.812 | **5.580** | 5.580 | 76.283 | **74.891** | 76.283 | 78.684 |
| | 1000 | 5.722 | 5.814 | 5.722 | 5.424 | 75.436 | 74.949 | 74.868 | 78.150 |
| | all | 6.056 | **5.704** | 6.056 | 5.920 | **72.554** | 74.683 | 73.790 | 79.467 |
| Gothic | 100 | 6.124 | 8.726 | 6.183 | 6.196 | 81.109 | 76.555 | 76.165 | 79.461 |
| | 500 | 6.694 | 8.708 | **6.565** | 7.091 | 76.484 | 76.502 | **72.869** | 75.740 |
| | 1000 | 6.934 | 8.701 | **6.676** | 6.904 | 75.244 | 76.537 | **72.391** | 76.874 |
| | all | 7.777 | 8.704 | **7.178** | 7.613 | 74.074 | 76.927 | **72.479** | 75.137 |
| Greek | 100 | 4.178 | 3.868 | 3.769 | 3.769 | 67.287 | 69.386 | 67.287 | 74.632 |
| | 500 | 4.071 | 3.868 | 3.949 | 3.949 | 66.323 | 69.403 | 66.323 | 74.886 |
| | 1000 | 4.190 | **3.868** | 3.985 | 3.985 | 66.847 | 69.369 | 66.847 | 74.801 |
| | all | 4.110 | **3.866** | 3.997 | 3.997 | 66.695 | 69.352 | 67.203 | 72.686 |
| Hebrew | 100 | 3.899 | 3.914 | **3.790** | 3.790 | 72.796 | 73.129 | 72.796 | 82.966 |
| | 500 | 3.928 | 3.912 | 3.828 | 3.828 | 76.625 | **73.114** | 76.625 | 76.316 |
| | 1000 | 3.904 | 3.908 | 3.859 | 3.859 | 73.724 | 73.145 | 73.724 | 78.131 |
| | all | 4.022 | 3.910 | 3.945 | 3.945 | 73.359 | 73.137 | 73.359 | 79.027 |

| Language | Training size | Perplexity | | | | Word Error Rate | | | |
|---|---|---|---|---|---|---|---|---|---|
| | | SL | SL bagging | MT-SL | MT-SL no proj. | SL | SL bagging | MT-SL | MT-SL no proj. |
| Hindi | 100 | 4.167 | 5.019 | 4.099 | 4.099 | 61.818 | 77.203 | 61.818 | 75.473 |
| | 500 | 4.015 | 5.018 | 4.015 | 4.150 | 60.958 | 77.197 | 60.958 | 76.208 |
| | 1000 | 4.235 | 5.016 | 4.235 | 4.557 | 62.850 | 77.197 | 62.850 | 73.142 |
| | all | 4.340 | 4.952 | 4.319 | 4.411 | 59.818 | 77.162 | 59.818 | 70.731 |
| Hungarian | 100 | 5.762 | 5.048 | **4.845** | 4.845 | 69.368 | 76.319 | 69.368 | 76.668 |
| | 500 | 5.727 | **5.047** | 5.184 | 5.184 | 68.949 | 76.319 | 68.599 | 80.091 |
| | 1000 | 5.801 | **5.051** | 5.215 | 5.215 | 68.809 | 76.249 | 68.914 | 79.776 |
| | all | 5.592 | 5.056 | 5.147 | 5.147 | 69.263 | 76.249 | 69.193 | 79.462 |
| Indonesian | 100 | 4.687 | 4.663 | **4.139** | 4.351 | 74.581 | 77.871 | 74.581 | 79.890 |
| | 500 | 4.630 | 4.658 | **4.296** | 4.296 | 71.411 | 77.880 | 71.411 | 78.204 |
| | 1000 | 4.601 | 4.663 | **4.494** | 4.494 | 70.901 | 77.807 | 70.901 | 78.520 |
| | all | 4.734 | 4.658 | **4.614** | 4.614 | 71.160 | 77.896 | 71.160 | 77.734 |
| Irish | 100 | 3.587 | **3.446** | 3.591 | 3.591 | 67.086 | 73.835 | 67.086 | 79.451 |
| | 500 | 3.548 | 3.453 | 3.463 | 3.463 | 66.230 | 73.835 | 66.230 | 73.659 |
| | 1000 | 3.594 | **3.453** | 3.559 | 3.559 | 66.885 | 73.886 | 66.885 | 76.077 |
| | all | 3.594 | **3.453** | 3.559 | 3.559 | 66.885 | 73.886 | 66.885 | 76.077 |
| Italian | 100 | 3.343 | 3.769 | 3.249 | 3.254 | 64.584 | 70.483 | **63.036** | 70.090 |
| | 500 | 3.450 | 3.765 | **3.276** | 3.276 | 58.054 | 69.688 | 58.054 | 63.867 |
| | 1000 | 3.466 | 3.761 | **3.355** | 3.355 | 57.880 | 69.094 | 57.880 | 62.757 |
| | all | 3.620 | 3.580 | 3.506 | 3.506 | 57.574 | 63.307 | 57.574 | 63.972 |
| Japanese | 100 | 2.981 | 3.586 | 2.888 | 3.162 | 64.651 | 82.419 | 64.651 | 74.700 |
| | 500 | 3.148 | 3.566 | 3.124 | 3.175 | 60.545 | 82.773 | 60.545 | 69.113 |
| | 1000 | 3.197 | 3.564 | 3.197 | 3.199 | 61.664 | 82.618 | 61.664 | 69.308 |
| | all | 3.196 | 3.538 | 3.221 | 3.255 | 61.837 | 82.946 | **59.632** | 73.626 |
| Latin | 100 | 4.707 | 6.825 | 4.707 | 5.137 | 79.580 | **74.320** | 75.690 | 79.660 |
| | 500 | 5.143 | 6.815 | 5.143 | 5.462 | 76.556 | **74.317** | 75.753 | 79.837 |
| | 1000 | 5.299 | 6.805 | 5.299 | 5.625 | 75.919 | 74.345 | 74.974 | 81.735 |
| | all | 6.241 | 6.711 | 6.239 | 6.428 | 75.179 | 74.300 | **72.662** | 78.787 |
| Norwegian | 100 | 5.089 | 6.190 | **4.911** | 4.911 | 74.544 | 74.435 | **72.514** | 76.952 |
| | 500 | 5.164 | 6.152 | **4.983** | 5.117 | 71.926 | 74.297 | **70.353** | 73.650 |
| | 1000 | 5.235 | 6.141 | 5.147 | 5.318 | 69.887 | 74.194 | 69.199 | 75.864 |
| | all | 5.733 | 6.009 | **5.632** | 5.882 | 69.487 | 72.937 | **66.716** | 75.626 |
| Old Church Slavonic | 100 | 5.959 | 11.445 | 5.886 | 6.203 | 74.092 | 77.813 | **72.284** | 74.588 |
| | 500 | 7.172 | 11.465 | **6.733** | 7.096 | 70.441 | 77.778 | 69.697 | 73.932 |
| | 1000 | 7.765 | 11.420 | **7.307** | 7.901 | 68.297 | 77.760 | 67.978 | 73.826 |
| | all | 8.889 | 11.086 | **8.465** | 8.949 | 68.067 | 75.970 | **66.968** | 69.981 |
| Persian | 100 | 3.231 | 3.489 | **3.012** | 3.209 | 67.501 | 72.542 | 61.352 | 61.352 |
| | 500 | 3.256 | 3.501 | 3.244 | 3.345 | 59.253 | 72.560 | 59.253 | 76.483 |
| | 1000 | 3.274 | 3.495 | 3.318 | 3.332 | 58.429 | 72.500 | **56.822** | 74.437 |
| | all | 3.339 | 3.473 | 3.339 | 3.423 | 58.164 | 72.554 | **55.354** | 65.353 |
| Polish | 100 | 4.686 | 8.871 | 4.745 | 5.276 | 65.963 | 69.578 | 65.078 | 70.210 |
| | 500 | 5.288 | 8.858 | **5.108** | 5.579 | 68.643 | 69.565 | **62.588** | 69.565 |
| | 1000 | 5.466 | 8.832 | **5.333** | 5.978 | 68.984 | 69.451 | **63.208** | 72.434 |
| | all | 6.404 | 8.295 | **6.184** | 6.865 | 63.802 | 68.466 | 63.802 | 69.363 |
| Portuguese | 100 | 3.761 | 4.359 | 3.679 | 3.891 | 74.216 | **67.880** | 72.819 | 79.643 |
| | 500 | 3.988 | 4.402 | 3.989 | 3.989 | 69.862 | 67.864 | 67.815 | 71.990 |
| | 1000 | **4.059** | 4.382 | 4.210 | 4.210 | 68.952 | 68.058 | 68.952 | 68.725 |
| | all | 4.288 | 4.308 | 4.342 | 4.342 | 68.757 | 67.490 | **65.491** | 69.829 |
| Romanian | 100 | 7.269 | 5.799 | **4.923** | 4.923 | 71.105 | 71.997 | 71.311 | 78.655 |
| | 500 | 7.269 | **5.792** | 6.288 | 6.288 | **69.526** | 71.997 | 70.899 | 79.753 |
| | 1000 | 7.269 | **5.792** | 6.288 | 6.288 | **69.526** | 71.997 | 70.899 | 79.753 |
| | all | 7.269 | **5.792** | 6.288 | 6.288 | **69.526** | 71.997 | 70.899 | 79.753 |
| Slovenian | 100 | 5.156 | 5.383 | **4.540** | 4.788 | 81.297 | **71.790** | 74.692 | 77.903 |
| | 500 | 5.429 | 5.385 | **4.993** | 4.993 | 71.231 | 71.790 | 71.231 | 80.213 |
| | 1000 | 5.387 | 5.384 | **5.083** | 5.083 | 71.588 | 71.783 | **68.754** | 77.257 |
| | all | 5.605 | 5.347 | 5.406 | 5.406 | 70.875 | 71.635 | **64.943** | 77.244 |
| Spanish | 100 | 3.109 | 3.125 | 2.983 | **2.969** | 67.607 | **66.355** | 67.607 | 76.468 |
| | 500 | 3.114 | 3.121 | 3.069 | 3.069 | 67.898 | **66.440** | 67.898 | 73.794 |
| | 1000 | 3.166 | 3.120 | 3.070 | 3.070 | 64.750 | 66.452 | 64.750 | 68.798 |
| | all | 3.265 | 3.107 | 3.176 | 3.176 | 64.702 | 66.136 | 64.702 | 71.727 |
| Swedish | 100 | 5.184 | 5.480 | **4.761** | 4.768 | 72.166 | 73.602 | 72.166 | 76.144 |
| | 500 | 5.379 | 5.482 | **5.108** | 5.108 | 70.268 | 73.634 | 70.268 | 77.311 |
| | 1000 | 5.482 | 5.481 | **5.202** | 5.202 | 68.971 | 73.662 | 68.971 | 75.125 |
| | all | 5.737 | 5.461 | 5.489 | 5.489 | 68.878 | 72.819 | 68.878 | 78.565 |
| Tamil | 100 | 8.289 | 6.884 | **5.859** | 5.859 | 66.999 | 76.102 | 66.098 | 74.395 |
| | 500 | 8.305 | 6.884 | **6.149** | 6.149 | 66.524 | 76.055 | 67.330 | 80.512 |
| | 1000 | 8.305 | 6.884 | **6.149** | 6.149 | 66.524 | 76.055 | 67.330 | 80.512 |
| | all | 8.305 | 6.884 | **6.149** | 6.149 | 66.524 | 76.055 | 67.330 | 80.512 |

Table 2: Detailed experimental results on the UNIDEP dataset when the 4 closest languages are used as related tasks. The closest languages used for each target task are reported in Table 3.

| Language | Training size | Perplexity | | | | Word Error Rate | | | |
|---|---|---|---|---|---|---|---|---|---|
| | | SL | SL bagging | MT-SL | MT-SL no proj. | SL | SL bagging | MT-SL | MT-SL no proj. |
| Ancient Greek | 100 | 4.053 | 4.960 | 3.998 | 4.288 | 82.445 | **77.187** | 81.998 | 81.998 |
| | 500 | 4.194 | 4.952 | 4.128 | 4.211 | 81.572 | 77.493 | 78.207 | 80.225 |
| | 1000 | 4.203 | 5.014 | 4.212 | 4.375 | 77.696 | 77.268 | 79.554 | 79.554 |
| | all | 4.582 | 4.651 | 4.612 | 4.640 | 75.596 | 76.220 | 76.850 | 76.850 |
| Arabic | 100 | 2.316 | 2.367 | 2.245 | 2.289 | 77.306 | **76.098** | 77.161 | 81.334 |
| | 500 | 2.298 | 2.365 | 2.285 | 2.285 | 68.804 | 75.839 | 68.804 | 77.468 |
| | 1000 | 2.315 | 2.362 | 2.287 | 2.287 | 68.066 | 75.970 | 68.066 | 80.260 |
| | all | 2.306 | 2.334 | 2.300 | 2.300 | 66.595 | 75.794 | 66.595 | 75.418 |
| Basque | 100 | 5.932 | 9.375 | 5.984 | 6.544 | 75.956 | 77.129 | **70.290** | 75.333 |
| | 500 | 6.057 | 9.307 | 5.961 | 7.264 | 76.300 | 76.892 | **65.820** | 73.121 |
| | 1000 | 6.261 | 9.271 | 6.170 | 7.281 | 77.129 | 76.747 | **68.582** | 72.586 |
| | all | 6.760 | 8.815 | 6.760 | 7.868 | 75.803 | 75.280 | **69.064** | 73.813 |
| Bulgarian | 100 | 5.103 | 6.853 | **5.001** | 5.596 | 74.647 | **67.056** | 70.861 | 73.288 |
| | 500 | 5.475 | 6.800 | 5.475 | 5.801 | 68.576 | 67.027 | 66.065 | 69.021 |
| | 1000 | 5.649 | 6.785 | **5.436** | 5.696 | 66.059 | 66.855 | **63.816** | 70.386 |
| | all | 6.162 | 6.403 | 6.162 | 6.269 | 66.018 | 66.303 | **64.024** | 67.596 |
| Croatian | 100 | 5.510 | 4.918 | **4.475** | 4.475 | 75.468 | **70.636** | 75.653 | 80.301 |
| | 500 | 5.336 | 4.927 | **4.608** | 4.608 | 77.503 | **70.636** | 75.561 | 75.561 |
| | 1000 | 5.454 | 4.896 | 4.841 | 4.841 | 77.526 | **70.636** | 76.069 | 76.069 |
| | all | 5.285 | **4.897** | 5.163 | 5.163 | 77.919 | **70.613** | 77.387 | 77.387 |
| Czech | 100 | 4.292 | 5.383 | 4.232 | 4.232 | 77.897 | 75.731 | 77.342 | 77.891 |
| | 500 | 4.482 | 5.373 | 4.482 | 4.518 | 77.257 | 75.155 | **72.426** | 74.847 |
| | 1000 | 4.530 | 5.370 | 4.531 | 4.548 | 75.254 | 75.451 | 74.346 | 75.049 |
| | all | 5.091 | 5.087 | 5.091 | 5.106 | 73.849 | 70.747 | 71.693 | 74.949 |
| Danish | 100 | 4.977 | 4.943 | **4.662** | 4.894 | 78.891 | **74.299** | 77.892 | 77.892 |
| | 500 | 5.033 | **4.929** | 5.045 | 4.963 | 71.737 | 74.041 | 72.430 | 78.005 |
| | 1000 | 5.189 | 4.952 | **4.828** | 4.828 | 71.737 | 73.816 | 71.737 | 77.409 |
| | all | 5.363 | **4.864** | 5.203 | 5.022 | 70.674 | 73.751 | 69.562 | 73.284 |
| Dutch | 100 | 6.456 | 6.777 | **5.956** | 5.956 | 77.525 | 77.307 | 76.470 | 80.807 |
| | 500 | 7.679 | **6.847** | 7.232 | 7.156 | 75.498 | 76.888 | 75.498 | 78.262 |
| | 1000 | 7.517 | **6.774** | 7.107 | 7.107 | 73.874 | 76.218 | 73.874 | 78.664 |
| | all | 8.025 | **6.509** | 8.062 | 8.062 | 72.785 | 73.020 | 72.098 | 77.608 |
| English | 100 | 5.203 | 7.387 | 5.115 | 5.444 | 73.853 | **70.478** | 73.956 | 73.956 |
| | 500 | 5.620 | 7.391 | 5.684 | 5.574 | 71.729 | 70.408 | 70.158 | 71.483 |
| | 1000 | 5.781 | 7.337 | **5.631** | 5.817 | 72.716 | 70.559 | 69.720 | 70.180 |
| | all | 6.464 | 6.456 | 6.681 | 6.438 | 67.626 | 70.754 | 67.262 | 71.284 |
| Estonian | 100 | 5.301 | 7.457 | 5.140 | **4.447** | 50.506 | 55.934 | 50.506 | 56.210 |
| | 500 | 6.108 | 7.864 | **5.250** | 5.250 | 49.586 | 53.450 | 49.770 | 57.038 |
| | 1000 | 6.605 | 7.433 | **6.011** | 6.374 | 50.138 | 52.346 | 50.138 | 57.590 |
| | all | 6.653 | 7.287 | 7.322 | **6.504** | 50.046 | 52.070 | 50.046 | 56.210 |
| Finnish | 100 | 5.598 | 9.683 | **4.777** | 4.777 | 68.774 | 73.697 | 68.595 | 71.606 |
| | 500 | 5.964 | 9.780 | 5.731 | **5.699** | 67.586 | 71.407 | 67.397 | 69.773 |
| | 1000 | 6.231 | 9.714 | **5.920** | 6.257 | 66.366 | 71.125 | 65.870 | 67.807 |
| | all | 7.709 | 8.494 | **7.516** | 7.672 | 63.811 | 66.091 | 63.558 | 64.885 |
| French | 100 | 3.744 | 3.825 | 3.643 | **3.540** | 69.245 | **62.247** | 65.787 | 68.548 |
| | 500 | 3.611 | 3.821 | 3.611 | 3.586 | 63.669 | 62.165 | 61.577 | 68.617 |
| | 1000 | 3.735 | 3.911 | **3.596** | 3.596 | **59.787** | 62.083 | 61.331 | 62.739 |
| | all | 3.823 | 3.736 | 3.800 | 3.800 | **59.732** | 62.097 | 60.894 | 64.858 |
| German | 100 | 5.756 | 5.829 | **5.392** | 5.392 | 78.498 | **73.685** | 78.458 | 80.968 |
| | 500 | 5.960 | 5.826 | 5.515 | **5.496** | 76.283 | **73.047** | 77.176 | 74.543 |
| | 1000 | 5.722 | 5.834 | **5.478** | 5.540 | 75.436 | 73.105 | 73.958 | 76.921 |
| | all | 6.056 | **5.651** | 6.056 | 6.117 | 72.554 | 72.728 | 72.264 | 76.979 |
| Gothic | 100 | 6.124 | 8.230 | **5.966** | 5.966 | 81.109 | 76.573 | 76.590 | 80.082 |
| | 500 | 6.694 | 8.261 | **6.539** | 6.539 | 76.484 | 76.077 | **73.348** | 76.023 |
| | 1000 | 6.934 | 8.334 | 6.876 | 6.876 | 75.244 | 76.147 | **72.018** | 75.031 |
| | all | 7.777 | 8.069 | **7.187** | 7.551 | 74.074 | 75.563 | **72.391** | 74.216 |
| Greek | 100 | 4.178 | 3.971 | **3.679** | 3.679 | 67.287 | 74.750 | 67.287 | 75.038 |
| | 500 | 4.071 | 3.975 | 3.961 | 3.961 | 66.323 | 74.344 | 66.323 | 75.207 |
| | 1000 | 4.190 | 3.959 | 3.997 | 3.997 | 66.847 | 73.549 | 66.847 | 72.821 |
| | all | 4.110 | 3.953 | 3.964 | 3.964 | 66.695 | 73.837 | 67.203 | 73.803 |
| Hebrew | 100 | 3.899 | 3.873 | **3.736** | 3.736 | 72.796 | 74.897 | 72.796 | 75.127 |
| | 500 | 3.928 | 3.873 | 3.802 | 3.802 | 76.625 | **74.889** | 76.625 | 78.702 |
| | 1000 | 3.904 | 3.873 | 3.910 | 3.910 | 73.724 | 74.881 | 73.724 | 77.957 |
| | all | 4.022 | 3.870 | 3.907 | 3.907 | 73.359 | 74.929 | 73.359 | 77.172 |

| Language | Training size | Perplexity | | | | Word Error Rate | | | |
|---|---|---|---|---|---|---|---|---|---|
| | | SL | SL bagging | MT-SL | MT-SL no proj. | SL | SL bagging | MT-SL | MT-SL no proj. |
| Hindi | 100 | 4.167 | 4.727 | 4.165 | 4.164 | 61.818 | 70.421 | 60.942 | 68.074 |
| | 500 | 4.015 | 4.713 | 4.003 | 4.144 | 60.958 | 70.108 | 60.271 | 65.886 |
| | 1000 | 4.235 | 4.709 | 4.202 | 4.202 | 62.850 | 70.297 | **61.001** | 66.746 |
| | all | 4.340 | 4.511 | 4.344 | 4.344 | 59.818 | 64.019 | 59.341 | 62.004 |
| Hungarian | 100 | 5.762 | 5.190 | **4.837** | 4.837 | 69.368 | 78.484 | 69.368 | 78.764 |
| | 500 | 5.727 | 5.121 | 5.136 | 5.136 | 68.949 | 78.589 | 67.971 | 78.449 |
| | 1000 | 5.801 | 5.215 | 5.118 | 5.118 | **68.809** | 79.008 | 70.031 | 76.808 |
| | all | 5.592 | 5.119 | 5.089 | 5.089 | 69.263 | 79.218 | 69.752 | 76.808 |
| Indonesian | 100 | 4.687 | 4.694 | **4.248** | 4.248 | 74.581 | 77.790 | 74.581 | 76.372 |
| | 500 | 4.630 | 4.672 | **4.339** | 4.339 | 71.411 | 78.382 | 71.411 | 77.985 |
| | 1000 | 4.601 | 4.675 | 4.522 | 4.522 | 70.901 | 78.439 | 70.901 | 76.745 |
| | all | 4.734 | 4.626 | 4.572 | 4.572 | 71.160 | 78.520 | 71.160 | 75.902 |
| Irish | 100 | 3.587 | 3.490 | 3.431 | 3.326 | 67.086 | 72.400 | 67.086 | 73.105 |
| | 500 | 3.548 | 3.457 | 3.417 | 3.417 | 66.230 | 72.425 | 66.230 | 74.969 |
| | 1000 | 3.594 | 3.455 | 3.425 | 3.425 | 66.885 | 72.098 | 66.885 | 72.450 |
| | all | 3.594 | 3.455 | 3.425 | 3.425 | 66.885 | 72.098 | 66.885 | 72.450 |
| Italian | 100 | 3.343 | 3.719 | 3.359 | 3.359 | 64.584 | 62.215 | **58.719** | 72.625 |
| | 500 | 3.450 | 3.717 | **3.336** | 3.336 | 58.054 | 62.320 | 58.054 | 66.594 |
| | 1000 | 3.466 | 3.678 | 3.457 | 3.457 | 57.880 | 61.822 | 56.962 | 65.274 |
| | all | 3.620 | 3.542 | 3.575 | 3.575 | 57.574 | 59.156 | 57.276 | 61.183 |
| Japanese | 100 | 2.981 | 3.492 | 3.003 | 3.030 | 64.651 | 67.524 | 64.651 | 69.963 |
| | 500 | 3.148 | 3.479 | 3.094 | 3.135 | 60.545 | 67.971 | 60.545 | 64.841 |
| | 1000 | 3.197 | 3.479 | 3.114 | 3.168 | 61.664 | 67.021 | 61.151 | 61.066 |
| | all | 3.196 | 3.274 | 3.243 | 3.243 | 61.837 | 66.008 | 61.837 | 63.632 |
| Latin | 100 | 4.707 | 6.542 | 4.713 | 4.972 | 79.580 | 78.638 | **75.902** | 81.986 |
| | 500 | 5.143 | 6.541 | 5.143 | 5.646 | 76.556 | 78.468 | 75.878 | 79.462 |
| | 1000 | 5.299 | 6.544 | 5.269 | 5.805 | 75.919 | 76.517 | 75.085 | 78.937 |
| | all | 6.241 | 6.407 | 6.235 | 6.526 | 75.179 | 77.213 | **73.198** | 79.170 |
| Norwegian | 100 | 5.089 | 6.234 | **4.929** | 4.929 | 74.544 | 71.695 | 72.483 | 72.483 |
| | 500 | 5.164 | 6.167 | 5.166 | 5.166 | 71.926 | 71.848 | **69.965** | 72.655 |
| | 1000 | 5.235 | 6.157 | 5.186 | 5.281 | 69.887 | 71.341 | **68.836** | 70.900 |
| | all | 5.733 | 5.800 | 5.743 | 5.739 | 69.487 | 70.738 | **68.132** | 70.550 |
| Old Church Slavonic | 100 | 5.959 | 10.250 | 5.894 | 5.899 | 74.092 | 71.735 | 72.656 | 75.917 |
| | 500 | 7.172 | 10.273 | 7.071 | **6.934** | 70.441 | 71.806 | **69.360** | 71.699 |
| | 1000 | 7.765 | 10.284 | 7.418 | **7.374** | 68.297 | 72.072 | 68.138 | 71.629 |
| | all | 8.889 | 9.712 | 8.662 | **8.523** | 68.067 | 71.505 | 68.262 | 70.370 |
| Persian | 100 | 3.231 | 3.536 | 3.158 | 3.161 | 67.501 | 74.311 | **63.272** | 70.124 |
| | 500 | 3.256 | 3.533 | 3.238 | 3.286 | 59.253 | 74.221 | 59.199 | 70.545 |
| | 1000 | **3.274** | 3.533 | 3.377 | 3.377 | 58.429 | 73.679 | **57.328** | 70.419 |
| | all | 3.339 | 3.523 | 3.335 | 3.418 | 58.164 | 72.843 | **56.100** | 67.585 |
| Polish | 100 | 4.686 | 10.034 | **4.387** | 5.003 | 65.963 | 63.220 | 65.963 | 68.984 |
| | 500 | 5.288 | 9.415 | **5.106** | 5.352 | 68.643 | 62.639 | 65.950 | 68.579 |
| | 1000 | 5.466 | 9.563 | **5.256** | 5.666 | 68.984 | 62.551 | 64.800 | 70.071 |
| | all | 6.404 | 8.088 | **6.048** | 6.537 | 63.802 | 60.389 | 63.802 | 69.742 |
| Portuguese | 100 | 3.761 | 4.448 | **3.631** | 3.827 | 74.216 | **63.704** | 73.696 | 74.427 |
| | 500 | 3.988 | 4.387 | 3.940 | 4.104 | 69.862 | **63.477** | 67.116 | 73.534 |
| | 1000 | 4.059 | 4.393 | 4.016 | 4.094 | 68.952 | **63.656** | 67.295 | 67.295 |
| | all | 4.288 | 4.382 | 4.113 | 4.195 | 68.757 | **63.672** | 65.053 | 67.912 |
| Romanian | 100 | 7.269 | 5.970 | **5.401** | 5.401 | 71.105 | 73.095 | 71.311 | 74.674 |
| | 500 | 7.269 | 5.961 | 5.936 | 5.936 | **69.526** | 74.262 | 70.899 | 80.851 |
| | 1000 | 7.269 | 5.961 | 5.936 | 5.936 | **69.526** | 74.262 | 70.899 | 80.851 |
| | all | 7.269 | 5.961 | 5.936 | 5.936 | **69.526** | 74.262 | 70.899 | 80.851 |
| Slovenian | 100 | 5.156 | 5.277 | **4.600** | 4.600 | 81.297 | 69.696 | 69.932 | 78.153 |
| | 500 | 5.429 | 5.267 | **5.026** | 5.026 | 71.231 | **69.945** | 71.231 | 76.920 |
| | 1000 | 5.387 | 5.289 | **5.055** | 5.055 | 71.588 | 69.629 | 69.804 | 72.713 |
| | all | 5.605 | 5.222 | 5.300 | 5.300 | 70.875 | 67.939 | 66.949 | 73.850 |
| Spanish | 100 | 3.109 | 3.079 | 3.015 | 3.015 | 67.607 | 64.045 | 67.607 | 68.688 |
| | 500 | 3.114 | 3.082 | 3.034 | 3.034 | 67.898 | 62.380 | 67.898 | 69.199 |
| | 1000 | 3.166 | 3.079 | 3.083 | 3.083 | 64.750 | 63.583 | 64.446 | 63.899 |
| | all | 3.265 | 3.062 | 3.140 | 3.140 | 64.702 | **61.213** | 62.514 | 62.514 |
| Swedish | 100 | 5.184 | 5.773 | **5.009** | 5.009 | 72.166 | **68.119** | 72.166 | 77.417 |
| | 500 | 5.379 | 5.708 | **5.225** | 5.225 | 70.268 | **68.360** | 70.268 | 76.056 |
| | 1000 | 5.482 | 5.773 | **5.277** | 5.277 | 68.971 | 68.286 | **67.230** | 75.218 |
| | all | 5.737 | 5.669 | **5.558** | 5.558 | 68.878 | 67.425 | 66.619 | 74.935 |
| Tamil | 100 | 8.289 | 6.988 | **6.334** | 6.334 | 66.999 | 76.339 | 66.098 | 78.284 |
| | 500 | 8.305 | 6.719 | **6.205** | 6.205 | 66.524 | 76.766 | 67.330 | 72.072 |
| | 1000 | 8.305 | 6.719 | **6.205** | 6.205 | 66.524 | 76.766 | 67.330 | 72.072 |
| | all | 8.305 | 6.719 | **6.205** | 6.205 | 66.524 | 76.766 | 67.330 | 72.072 |

| Target task | 4 closest tasks w.r.t. subspace distance (closest first) | | | |
| --- | --- | --- | --- | --- |
| Ancient Greek | Old Church Slavonic | Latin | Gothic | Hungarian |
| Arabic | Czech | Polish | Persian | Slovenian |
| Basque | Finnish | Polish | Czech | Indonesian |
| Bulgarian | Czech | Norwegian | Finnish | Slovenian |
| Croatian | Estonian | Slovenian | Czech | Finnish |
| Czech | Finnish | Norwegian | Bulgarian | Danish |
| Danish | Norwegian | Swedish | English | Czech |
| Dutch | German | Norwegian | Danish | English |
| English | Norwegian | Danish | Italian | Swedish |
| Estonian | Finnish | Swedish | Norwegian | Polish |
| Finnish | Estonian | Czech | Swedish | Norwegian |
| French | Italian | Spanish | German | English |
| German | Dutch | Swedish | English | French |
| Gothic | Old Church Slavonic | Latin | Ancient Greek | Finnish |
| Greek | Swedish | Spanish | Czech | German |
| Hebrew | Portuguese | Norwegian | Czech | Danish |
| Hindi | Japanese | Croatian | Tamil | Persian |
| Hungarian | Danish | Ancient Greek | German | Portuguese |
| Indonesian | Finnish | Czech | Bulgarian | Norwegian |
| Irish | Polish | Czech | Greek | Arabic |
| Italian | English | French | Spanish | Dutch |
| Japanese | Hindi | Persian | Arabic | Tamil |
| Latin | Old Church Slavonic | Ancient Greek | Gothic | Finnish |
| Norwegian | Danish | English | Swedish | Czech |
| Old Church Slavonic | Latin | Gothic | Ancient Greek | Finnish |
| Persian | Japanese | Czech | Swedish | Finnish |
| Polish | Slovenian | Czech | Finnish | Estonian |
| Portuguese | Hebrew | Norwegian | Italian | Danish |
| Romanian | Finnish | Estonian | Norwegian | Czech |
| Slovenian | Polish | Czech | Danish | Swedish |
| Spanish | French | Italian | Portuguese | Greek |
| Swedish | Danish | Norwegian | Finnish | Estonian |
| Tamil | Finnish | Indonesian | Basque | Croatian |

Table 3: Related tasks used in the UNIDEP experiment. The 4 closest tasks were selected using subspace distance (i.e. Frobenius norm of the difference between the orthogonal projection matrices) between the space spanned by the top 50 left singular vectors of their Hankel matrices. The common basis of prefixes/suffixes $(\mathcal{P}, \mathcal{S})$ for these Hankel matrices was obtained by taking the union of the 100 most frequent prefixes/suffixes for each task.