[Reviews · NeurIPS 2017]

Reviewer 1



Multitask Spectral Learning of Weighted Automata The paper proposes an approach to estimate a common representation of multiple weighted finite automata (WFA), via multitask spectral learning. To do so, the multitask problem is formulated as novel vector-valued WFAs, from which a joint feature space is extracted using spectra learning. The method is evaluated in synthetic (randomly generated WFAs) and real data (task-related learning using 33 languages), where the multitask approach is shown to outperform single task learning. This paper appears to be a first attempt to the multitask approach for WFAs. One major comment is perhaps a somewhat weaker description of the context and motivation. The approach is immediately presented with a spectral learning (SL) of WFAs, without spending too much time on when, where and why SL-WFA is used, in what context - nor on referencing the advantages and limitations of other competing algorithms in that context. A small related work section is provided, but may not be enough convincing on why the proposed approach is better. The evaluation is also comparing SL-WFAs against (M)SL-WFAs. This is nice to evaluate single vs multi-task, but give no information on why one should use the proposed approach, ML-SL, over other methods. The paper, however, presents a nice methodology for estimating a joint feature maps, with potential applications where finite-state machines are used (language-related tasks are illustrated). The methodology is well supported with upper bound conditions and an analysis on computational complexity.

Reviewer 2



In this paper, the authors have proposed to estimate m weighted automata based on spectral learning algorithms applied to the associated Hankel tensor. Specifically, they apply rank factorization via SVD to the Hankel Tensor. They also how this factorization can then be used to learn individual functions. I have the following comments: Is the factorization technique proposed here for Hankel tensor a novel contribution of this paper? In Definition 2, the authors introduce a measure of relatedness tau. Is there any specific reason for this specific definition? I wasn't able to relate it to other quantities in the paper; more detail on the purpose of defining this quantity would be very helpful. Would it dictate the choice R - the common rank? I was hoping that the benefit of estimating multiple functions together would be apparent through dependence of error measures on tau in Theorem 5 (maybe in a more general version of it) or computational complexity. Specifically, it seems the computational complexity is worse than when each individual WFAs are estimated separately. Can the relatedness not be leveraged in any way? I certainly feel that it would be very interesting and illuminating to see a more general version of Theorem 5. If the WFAs are minimally related - would the performance be worse than doing it individually? Specially, in simulation what if d_S were 0 or at least less than d_T. It seemed concerning that even when there is commonality - if d_S and d_T are same MT-SL seems to offer no benefit compare to simple SL. In showing the benefit of multiple learning at least for maximally related WFAs, the authors note that the estimation error for enough tasks would be O(T^2) compared to O(T) if done individually - where T is || E ||_F/s_R(H). I couldn't follow why O(T^{2}) would be better that O(T); is T <= 1?

Reviewer 3



SUMMARY The paper studies the problem of multitask learning of WFAs. It defines a notion of relatedness among tasks, and designs a new algorithm that can exploit such relatedness. Roughly speaking, the new algorithm stacks the Hankel matrices from different tasks together and perform an adapted version of spectral learning, resulting in a vv-WFA that can make vector-valued predictions with a unified state representation. A post-processing step that reduces the dimension of the WFA for each single task is also suggested to reduce noise. The algorithm is compared to the baseline of learning each task separately on both synthetic and real-world data. COMMENTS Overall this is a well written paper. However, I do have a concern in the experiment section: it is important to compare to the baseline where all data from different tasks are bagged together and treated as if they came from the same task. At least when all the tasks are the same, this should outperform everyone else as it makes full use of all the data. Of course, when the tasks are not related, such a practice may lead to asymptotic approximation error, but how large is this error practically? If this error is small on the datasets used in the experiment section, then such datasets are not interesting as any algorithm that does some kind of data aggregation would show improvement over single-task learning. If possible I would like to see some results (even if they are primary) on this comparison during rebuttal. It would be good to also compare to Alg 1 without the projection step to see how much improvement this post-processing procedure brings. The paper's presentation may be improved by discussing the application scenario of multi-task learning of WFAs. As a starter, one could consider natural language modeling tasks where we need to make predictions in different contexts (e.g., online chat vs newspaper articles) and have access to datasets in each of them. In this example, it is natural to expect that basic grammar is shared across the datasets and can be learned together. Of course, one can always aggregate all datasets into a big one and build a single model (which corresponds to the baseline I mentioned above), and the disadvantage is that the model cannot leverage the context information available at prediction phase. Two additional suggestions: - The current algorithm implicitly assumes equal weights among all the tasks. This should work well when the size of the datasets are roughly the same across tasks, but when they differ a lot I suspect that the algorithm could misbehave. In this case you might want to consider a weighted approach; see Kulesza et al, Low-Rank Spectral Learning with Weighted Loss Functions. - Here is another reason for doing the projection step: consider the case when the m tasks are completely unrelated, and each of them requires n states. Single-task learning would need n*m^2 parameters for each character in the alphabet, while the multi-task learning uses a model of size (nm)^2. The projection step eliminates such redundancy. MINOR ISSUE Line 93: as far as I know, it is not required that empty string is included in prefixes or suffixes. (At least this is true in the PSR literature which I am more familiar with.) The author(s) might want to double check on this. ============================== Thanks for the rebuttal and the additional results. No complaints! Will keep arguing for acceptance.

Reviewer 4



This paper presents a natural extension of spectral methods for learning WFAs to the multitask setting. The idea is quite simple, one can jointly train WFAs that share the same underlying operators but that have different ending vectors. The interpretation of this as sharing a latent (forward) space among different tasks seems sound. I found the paper self contained, clear and easy to read. Although the idea is quite simple if one is familiar with spectral learning of WFAs, I am quite confident it is novel. I believe this is a nice contribution. My one criticism is that the proposed notion of relatedness might be too strong to be useful in real learning problems. It would be nice if the authors could discuss ways in which this notion of relatedness could be relaxed.